# The activation of IgM- or isotype-switched IgG- and IgE-BCR exhibits distinct mechanical force sensitivity and threshold

**Zhengpeng Wan[1], Xiangjun Chen[1], Haodong Chen[2], Qinghua Ji[3], Yingjia Chen[1], Jing Wang[1], Yiyun Cao[1], Fei Wang[2], Jizhong Lou[3], Zhuo Tang[2], Wanli Liu[1]\***

[1]MOE Key Laboratory of Protein Sciences, Collaborative Innovation Center for Diagnosis and Treatment of Infectious Diseases, School of Life Sciences, Tsinghua University, Beijing, China; [2]Natural Products Research Center, Chengdu Institution of Biology, Chinese Academy of Science, Chengdu, China; [3]Laboratory of RNA Biology, Institute of Biophysics, Chinese Academy of Sciences, Beijing, China

**Abstract** B lymphocytes use B cell receptors (BCRs) to sense the physical features of the antigens. However, the sensitivity and threshold for the activation of BCRs resulting from the stimulation by mechanical forces are unknown. Here, we addressed this question using a double-stranded DNA-based tension gauge tether system serving as a predefined mechanical force gauge ranging from 12 to 56 pN. We observed that IgM-BCR activation is dependent on mechanical forces and exhibits a multi-threshold effect. In contrast, the activation of isotype-switched IgG- or IgE-BCR only requires a low threshold of less than 12 pN, providing an explanation for their rapid activation in response to antigen stimulation. Mechanistically, we found that the cytoplasmic tail of the IgG-BCR heavy chain is both required and sufficient to account for the low mechanical force threshold. These results defined the mechanical force sensitivity and threshold that are required to activate different isotyped BCRs.

**\*For correspondence:** liuwanli@ biomed.tsinghua.edu.cn

**Competing interests:** The authors declare that no competing interests exist.

## Introduction

B lymphocytes are responsible for the protective antibody responses arising from the recognition of the pathological antigens by the surface expressed B cell receptor (BCR) (*Kurosaki et al., 2010*). The BCR is composed of a membrane-bound immunoglobulin (mIg) and a non-covalently associated heterodimer of Igα and Igβ in a 1 mIg: 1 Igα–Igβ heterodimer stoichiometry (*Schamel and Reth, 2000*; *Tolar et al., 2005*). BCRs are distinguished from other types of receptors by their ability to recognize a wide range of antigen molecules. In addition to the BCR's ability to recognize antigen diversity, the activation of BCR signaling is also efficiently regulated by the presentation of variable forms of antigens. These forms include antigen density (*Fleire et al., 2006*; *Liu et al., 2010a*), antigen affinity (*Fleire et al., 2006*; *Liu et al., 2010a*), antigen valency (*Bachmann et al., 1993*; *Liu et al., 2004*; *Liu and Chen, 2005*), the Brownian mobility feature of the antigen (*Wan and Liu, 2012*), and the stiffness feature of the substrates presenting the antigen (*Wan et al., 2013*; *Zeng et al., 2015*). All these results suggest that the BCR is an extraordinary receptor which can efficiently discriminate the chemical and physical features of an antigen ligand.

Numerous early studies have investigated how the chemical cues from the antigen determine the strength of the signaling cascade mediated by the BCR (*Harwood and Batista, 2010*; *Pierce and Liu, 2010*). However, chemical cues are not the only type of external information that is delivered to the BCRs by antigens. In fact, physical cues are another layer of important information derived from the antigens for the purposes of regulating B cell activation and subsequent responses (*Liu et al., 2015*). For example, a recent study by Tolar and his colleagues demonstrated that B cells used the mechanical forces to rupture the bonds between BCRs and membrane-bound antigens. The authors

**eLife digest** The immune system protects us from invading bacteria and other microbes. Immune cells called B cells help the immune system to identify the microbes so they can be destroyed. These cells make proteins called antibodies that bind to molecules from the microbes known as antigens. The B cells only start to produce antibodies when they bind to a specific antigen via a large group or 'complex' of proteins on the surface of the B cell called the 'B cell receptor'. After the body has defeated the microbe, some of the B cells will become memory B cells, which are primed to rapidly respond to the antigen if the same microbe tries to invade again in future.

Previous work on the B cell receptor has largely focused on the chemical features of the antigens. However, recent research suggests that B cell receptors are also influenced by physical cues from the antigen. For example, the stiffness of the surface of the host cell or microbe that is displaying the antigen may exert a mechanical force on the B cell receptor as it binds to the antigen. However, it is not clear what role these mechanical forces play in triggering B cell activation and antibody production.

Before a B cell encounters an antigen, it expresses a type of B cell receptor called the IgM-BCR, but memory B cells express different types of B cell receptors. Wan et al. investigated how the different B cell receptors are activated using a technique involving a DNA-based tension gauge. The experiments show that the activation of IgM-BCRs depends on the amount of mechanical force applied. Low levels of mechanical force only weakly activated the receptors, while higher levels of force resulted in more robust activation.

In contrast, small amounts of mechanical force were sufficient to strongly activate the other two types of B cell receptors, IgG-BCR and IgE-BCR, on memory B cells. This may help memory B cells to be activated more quickly than other B cells that haven't encountered an antigen before. The next challenge is to understand why the B cell receptors on memory B cells are less dependent on mechanical forces than IgM-BCRs.

found that only the high affinity BCR and antigen microclusters would be internalized for antigen processing, giving a new mechanism to explain B cell affinity discrimination (*Natkanski et al., 2013*). Physiologically, the physical cues can also come from the stiffness of the substrates presenting the antigens. Stiffness usually describes the extent that an object can resist deformation in response to an applied force. Stiffness is quantified by Young's modulus with a measurable unit of Pascal (Pa or N/m$^2$ or m$^{-1}$·kg·s$^{-2}$) (*Discher et al., 2005*). B cells could encounter the antigens presented on substrates with various levels of stiffness in vivo (*Bachmann and Jennings, 2010*). For example, viral capsid substrates presenting antigen exhibited a high degree of stiffness (45,000–1,000,000 kPa) (*Mateu, 2012*), while substrates presenting antigen on the membrane of the host cell infected by virus showed a medium level of stiffness (0.01–1000 kPa) (*Nemir and West, 2010*). Additionally, B cells can also efficiently acquire antigens that are associated with extracellular matrix (ECM) in the tissue (*Ciechomska et al., 2014*), which is well documented to exhibit an extraordinary range of stiffness from 0.012 to 20 kPa (*Bao and Suresh, 2003*; *Engler et al., 2004*; *Paszek et al., 2005*; *Nemir and West, 2010*). Soluble antigen in plasma has also been detected and displayed a remarkably low level of stiffness (several Pa) (*Araujo Gde et al., 2012*). More recently, Kam and his colleagues quantified the traction forces between T cell receptors (TCRs) and the pillar substrates presenting antigens (*Bashour et al., 2014*). Indeed, significant shape changes of the substrates were observed due to the bending of the pillars by TCRs after TCR and antigen recognition (*Bashour et al., 2014*). Since stiffer substrates are known to be more resistant to shape changes (*Trappmann et al., 2012*), it is reasonable to expect that antigens presented on stiffer or softer substrates will inevitably produce and deliver different mechanical forces to the BCRs. Indeed, our recent study showed that the activation of a B cell is very sensitive to the stiffness of the substrates presenting the antigens (*Wan et al., 2013*; *Zeng et al., 2015*).

Thus, all these studies suggest that physical cues, such as mechanical force, comprise important external information delivered to the BCRs by antigens for the purposes of regulating B cell activation and subsequent responses. However, the mechanism regulating the strength of the BCR activation in response to mechanical forces that are delivered to the BCR by the antigens on the grounding

substrates remains unknown (*Pierce and Liu, 2010*). There are several important questions which need to be addressed. First of all, does the BCR itself have mechanosensing capability, or do B cells perform mechanosensing through the conventional mechanosensors such as lymphocyte function-associated antigen 1 (LFA-1) (*Chen et al., 2010*, *2012*). Second, is there a threshold for BCR activation from the mechanical forces that are delivered to the BCR by the antigen? Third, how sensitive will the BCR be toward different mechanical forces above that threshold in the initiation of BCR activation? Fourth, B cells use different isotypes of BCRs to recognize antigens and initiate transmembrane activation signaling (*McHeyzer-Williams and McHeyzer-Williams, 2005*; *Pierce and Liu, 2010*). Mature naive B cells use IgM-BCRs, while memory B cells mainly use IgG-BCRs along with a small fraction that use IgE-BCRs. We do not know how IgM-BCRs expressed by naive B cells and IgG-BCRs (or IgE-BCRs) expressed by memory B cells show differences in terms of the sensitivity and threshold toward mechanical forces in their activation.

A major obstacle to answer all of these questions is that it is technically challenging to set up an experimental system with predefined mechanical forces between BCR and antigen, which shall then be linked to an assay to accurately measure BCR activation. This is especially challenging since it is known that BCR activation may begin seconds after the recognition of the BCR and the antigen (*Pierce and Liu, 2010*). To overcome these technical difficulties, we took advantage of a platform utilizing a dsDNA-based tension gauge tether (TGT) that was recently developed by Ha and his colleagues (*Wang and Ha, 2013*). DNA is an excellent molecule showing force application geometry that has been accurately calculated and validated by in vitro single molecule force measurements (*Albrecht et al., 2003*; *Lang et al., 2004*). Ha and his colleagues used these TGT sensors to probe the tension on a single integrin–ECM ligand (cyclic RGDfK) bond required for cell adhesion. Here, we modified this TGT system by conjugating the B1-8 BCR-specific antigen, 4-Hydroxy-3-nitrophenylacetyl (NP), to the ligand chain of the TGT system (NP-TGT). By doing so, we acquired a series of 8 TGT sensors with predefined mechanical force of 12, 16, 23, 33, 43, 50, 54, and 56 pN respectively. We analyzed the activation of BCRs in response to these TGT molecules using high resolution high speed live cell imaging techniques via total internal reflection fluorescence microscopy (TIRFM). This allowed us to determine the sensitivity and threshold for the mechanical force signal in the activation of IgM-BCR or isotype-switched IgG-BCR and IgE-BCR.

## Results

### The construction of B1-8-BCR-specific NP-TGT mechanical force sensors

To further our understanding of how mechanical forces influence BCR activation, we constructed the NP-TGT sensors by modifying a dsDNA-based TGT system (*Wang and Ha, 2013*). Each NP-TGT molecule is composed of two single-stranded DNA (ssDNA) molecules with different modifications (*Figure 1A,B*). The first ssDNA molecule is biotin-conjugated at different positions to provide a defined range of rupture force anchoring positions as illustrated in *Figure 1B*. In the original version of the TGT system utilized by Ha and his colleagues (*Wang and Ha, 2013*), the second ssDNA molecule was conjugated with a well-characterized integrin ligand, cyclic RGDfk peptide, to provide an integrin binding site for quantifying the mechanical force spectrum (12, 16, 23, 33, 43, 50, 54, and 56 pN) in the activation of integrin molecules. As stated by the authors (*Wang and Ha, 2013*), TGT molecules with modifications can provide an experimental system for the study of many other types of receptors. Here, we conjugated the B1-8-BCR-specific antigen, NP, to the second ssDNA molecule (*Figure 1A,B*, *Figure 1—figure supplement 1A*). Since we just exchanged the cyclic RGDfk peptide with the NP hapten antigen and did not change the sequence and design of these dsDNA-based TGT sensors, our NP-TGT system shall have the same tension gauge scales as the one used by Ha and his colleagues (*Wang and Ha, 2013*). In our experimental system, each NP-TGT molecule can be recognized by the B1-8-IgM-BCRs (*Figure 1A*). Thus, the NP-TGT sensor that is immobilized on the surface of coverslip would be ruptured if the mechanical force applied by the B1-8-IgM-BCR is larger than the predefined tension force of a certain NP-TGT sensor. A key feature of dsDNA-based mechanical force sensor is that the specific force value of each sensor is not an absolute value but presents a distribution with a full width at half maximum (FWHM) of 5 pN for unzipping rupture mode and 15 pN for shearing rupture mode (*Lang et al., 2004*). So, precisely, the actual range of any given NP-TGT sensor represents a distribution with the most possible value of rupture force at 12, 16, 23, 33, 43, 50, 54, and 56 pN, respectively.

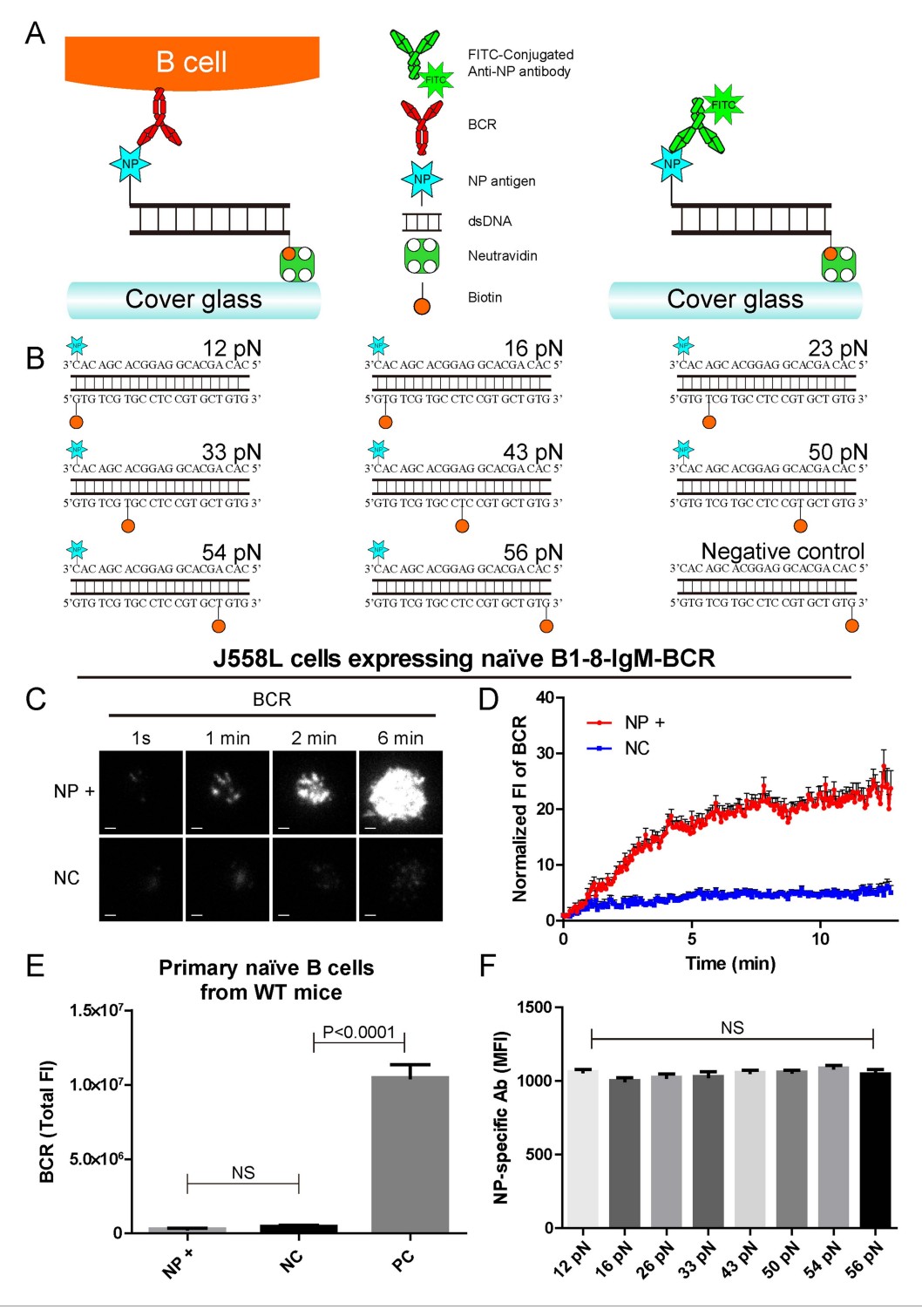

**Figure 1**. The construction of B1-8-BCR-specific NP-TGT mechanical force sensor system. (**A**) Schematic representation of the NP-TGT and NP-specific B1-8-BCR expressing B cells. NP-TGT molecule is immobilized on the surface of coverslip, which will get ruptured if the mechanical force applied by the B1-8-BCR is larger than the predefined tension force of a certain NP-TGT (for example 56 pN is depicted in the figure). FITC-conjugated anti-NP antibody is used to quantify the molecule density of each different type of NP-TGT sensors tethered on coverslip. (**B**) The dsDNA geometries and predefined tension force of all eight NP-conjugated TGT sensors and one control TGT without NP conjugation. (**C**) Representative TIRFM images showing the dynamics of the synaptic accumulation

*Figure 1. continued on next page*

*Figure 1. Continued*

of BCRs from J558L cells expressing B1-8-IgM-BCR in contact with coverslip presenting 56 pN NP-TGT sensor or control TGT (NC) at the indicated time points. Scale bar is 1.5 μm. (**D**) Comparisons of averaged traces showing the dynamic accumulation of BCRs as demonstrated in (**C**) in a 13 min TIRFM imaging time course. Bars represent mean ±SEM. Data were from at least 20 cells over three independent experiments. (**E**) Primary mature naive B cells from wild-type C57BL/6 mice expressing non-NP-specific IgM-BCR did not initiate the activation when encountering 56 pN NP-TGT sensor compared to the response of the same B cells encountering 56 pN TGT sensor without NP conjugation. Biotin-conjugated goat anti-mouse IgM surrogate antigens were used as a positive control to efficiently drive the synaptic accumulation of IgM-BCRs in B cell activation. Bars represent mean ±SEM. Two-tailed *t* tests were performed for the statistical comparisons. Data were from at least 30 cells over three independent experiments. (**F**) Quantification of the mean fluorescence intensity (MFI) of FITC-conjugated NP-specific antibodies on the surface of coverslip tethering the same amount of NP-TGT sensors. Bars represent mean ±SEM. Two-tailed *t* tests were performed for the statistical comparisons. The surface density is 29.0 molecule/μm², seeing more in *Figure 2—figure supplement 1*.

The following figure supplement is available for figure 1:

**Figure supplement 1**. The quality control of NP-TGT sensor based experimental system.

---

We first determined whether the NP-TGT sensor can trigger the activation of the B cells expressing B1-8-IgM-BCR. We used a similar protocol as reported (*Wang and Ha, 2013*) to tether the highest force NP-TGT molecule (mean rupture force 56 pN in the condition they used) with NP-conjugated ssDNA on coverslip pre-coated with neutravidin. We also used the same TGT molecule without NP conjugation as a negative control (NC) (*Figure 1B*). DyLight 649 AffiniPure Fab Fragment Goat Anti-Mouse IgM, μ chain specific antibodies was used to pre-label the B1-8-IgM-BCRs on J558L cells (J558L cells expressing B1-8-IgM-BCR) before TIRFM imaging experiment as reported in our previous studies (*Liu et al., 2010a*). We found that J558L cells expressing B1-8-IgM-BCR initiated the activation responses as quantified by the dramatic accumulation of BCRs into the contact interface of B cells with coverslip presenting 56 pN NP-TGT sensor and formed a typical B cell immunological synapse (IS) as illustrated by the time lapse TIRFM images (*Figure 1C,D*, *Video 1*). These results were not observed with the negative control (NC) 56 pN TGT without NP conjugation (*Figure 1C,D*, *Video 1*). Additional experiments showed that neutravidin would not dissociate from the coverslip during the 10-min time course of our experiments (*Figure 1—figure supplement 1B,C*), and NP-TGT can only be attached to coverslip in a neutravidin-dependent manner (*Figure 1—figure supplement 1D*). Further experiments also indicated that the non-specific attachment of NP-TGT molecules if any on the coverslip without pre-coated neutravidin cannot induce the synaptic accumulation of BCRs (*Figure 1—figure supplement 1E*).

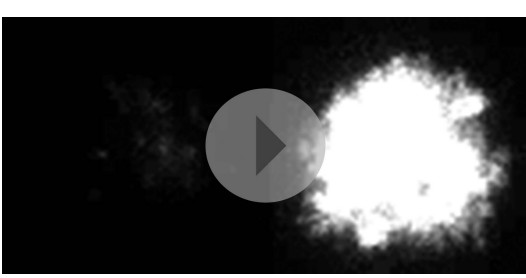

**Video 1.** Time lapse images showing the dynamics of the synaptic accumulation of BCRs from J558L cells expressing naive B1-8-IgM-BCR in contact with coverslip presenting 56 pN NP-TGT or control TGT (NC) sensor. Scale bar is 1.5 μm. The video was recorded with a 4-s time interval and is shown at 30 frames per second. Related to *Figure 1C*.

Since TGT is a dsDNA-based molecule, we were concerned that dsDNA may directly stimulate B cells in a BCR-independent manner. To check it, we examine the NP-TGT-triggered responses of primary mature naive B cells from wild-type C57BL/6 mice, which do not express NP-specific B1-8-IgM-BCRs (*Figure 1E*). These primary B cells did not accumulate BCRs into the B cell IS in response to coverslip presenting NP-TGT sensor compared to the case of coverslip alone, although a dramatic accumulation was readily observed in the case of encountering coverslip that was coated with anti-mouse IgM surrogate antigens as a positive control (*Figure 1E*). The B cell activation is known to be very sensitive to the density of antigen on the surface of the antigen-presenting substrates (*Fleire et al., 2006*; *Liu et al., 2010a*). To exclude such

a variant, we then used NP-specific antibodies to quantify the density of each NP-TGT sensor (*Figure 1F*). Thus, we established the B1-8-BCR-specific NP-TGT sensor system which showed a predefined mean mechanical force gauge ranging from 12 to 56 pN.

## The synaptic accumulation of the IgM-BCRs is dependent on mechanical forces and exhibits a multi-threshold effect

We compared the response of naive B1-8-IgM-BCR expressing B cells after immune recognition of these series of NP-TGT molecules. Numerous early studies showed that after BCR-antigen recognition, B cells immediately begin to spread over the antigen-containing surfaces to form the B cell IS and attempt to acquire the antigens by actively accumulating BCRs and antigens into the B cell IS (*Fleire et al., 2006*; *Liu et al., 2010a, 2010b, 2010c*; *Seeley-Fallen et al., 2014*). Thus, we first examined the B cell spreading response and the concomitant accumulation of BCRs into the B cell IS by TIRFM imaging using J558L cells expressing B1-8-IgM-BCR in NP-TGT-based experimental system (*Figure 2A*). The results suggested that NP-TGT sensors with higher mechanical force generally triggered more aggressive spreading response and enhanced BCR synaptic accumulation than NP-TGT molecules with lower mechanical force (*Figure 2A–C*). More strikingly, a careful examination of the response of B cells toward these series of NP-TGT sensors with mechanical forces ranging from 12 to 56 pN suggested that there are three levels of thresholds. The low-force (12–16 pN) NP-TGT sensors triggered a weak activation, whereas the middle-force (23–43 pN) NP-TGT sensors initiated a medium-level activation, and the high-force (50–56 pN) NP-TGT sensors accounted for a strong activation (*Figure 2A*). This unique pattern of the dependence on mechanical forces suggested that IgM-BCR activation exhibits a multi-threshold effect. Specifically, an increase of the mechanical force from one threshold to another threshold dramatically enhanced the activation of the naive IgM-BCRs. However, within these three threshold barriers, increasing the mechanical force, such as from 23 pN to 33 pN or to 43 pN (the medium level threshold), will not result in dramatically enhanced activation responses of IgM-BCRs.

We also validated these observations by analogously examining the activation response of primary mature naive B cells expressing B1-8-IgM-BCR from IgH$^{B1-8/B1-8}$ Igk$^{-/-}$ transgenic mice (referred to as primary naive B cells expressing B1-8-IgM-BCR) (*Hauser et al., 2007*) upon the recognition of these NP-TGT sensors (*Figure 2B*, *Figure 2—figure supplement 1A*). Furthermore, we were concerned that the density of the NP-TGT sensors on coverslip might affect the pattern of the activation response of the naive IgM-BCRs. Thus, we quantified the density (number of molecules/$\mu m^2$) of NP-TGT sensors on the coverslip, which was coated with different incubation concentration (2, 5, 10, 50 nM) of NP-TGT sensor (*Figure 2—figure supplement 1B–E*) following a published protocol (*Liu et al., 2010a*; *Wang and Ha, 2013*). At the density of 4.0 NP-TGT molecule/$\mu m^2$ (5 nM incubation concentration), we can capture the same pattern of the activation response of IgM-BCRs (*Figure 2—figure supplement 1F,G*). At the very low density of 0.3 NP-TGT molecule/$\mu m^2$ (2 nM incubation concentration), we found that only 56 pN NP-TGT sensor can very mildly trigger the BCR accumulation, while both 12 pN and 43 pN NP-TGT failed to trigger B cell activation (data not shown), suggesting that we cannot use the incubation concentration of 2 nM for these experiments. Thus, it is clear that the activation of the IgM-BCR is extremely sensitive to the changes in mechanical forces exceeding the thresholds such as the increase in the forces from 12 pN to 23 pN instead of 16 pN, or from 23 pN to 56 pN instead of 43 pN. In the following experiments, we chose the 12 pN, 43 pN, and 56 pN NP-TGT molecules in each of the three threshold barriers (low, medium, and high) for further analyses.

To investigate the temporal dependence of BCR accumulation into the B cell IS on mechanical forces, we took advantage of high speed time lapse TIRFM imaging to test the dynamics of BCR accumulation into the B cell IS and the growing size of contact area starting with the initial immune recognition of BCRs with 12 pN, 43 pN, and 56 pN NP-TGT sensors (*Figure 2C–E*, *Video 2*). As expected, we found that the synaptic accumulation of the BCRs in the entire 10-min time lapse was dependent on mechanical forces with high-force NP-TGT sensor (56 pN) exhibiting the highest accumulation, medium force NP-TGT sensor (43 pN) showed medium-level accumulation, and low-force NP-TGT sensor (12 pN) had the lowest level of accumulation, consistent with the results from the above mentioned end point experiments. Using these NP-TGT molecules, we also showed that there were no obvious internalization during the 10-min time course of our experiments (*Figure 2—figure supplement 2A–C*), consistent with the published studies (*Fleire et al., 2006*;

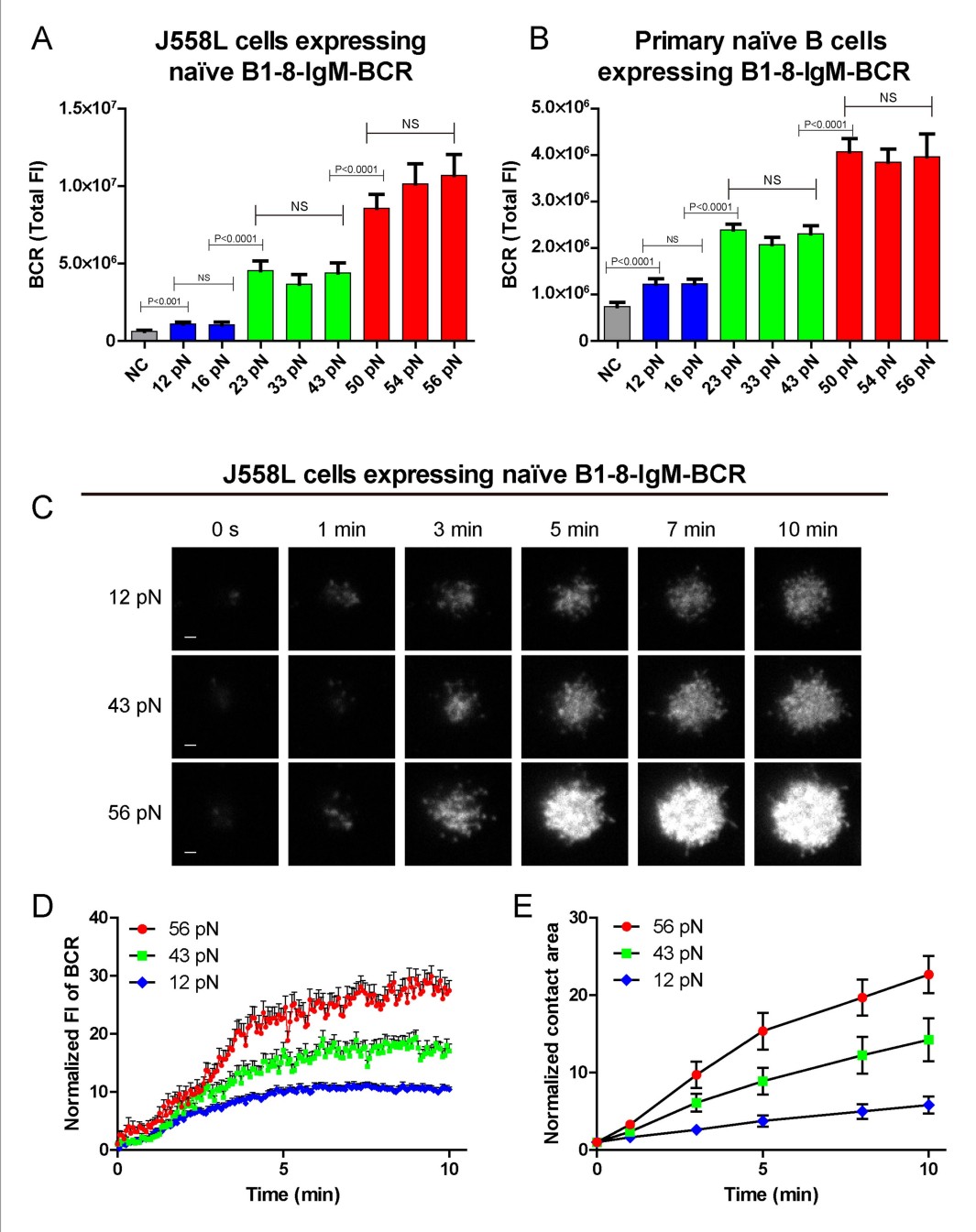

**Figure 2**. The synaptic accumulation of the IgM-BCRs is dependent on mechanical forces and exhibits a multi-threshold effect. (**A**, **B**) Statistical quantification of the synaptic recruitment of IgM-BCR in J558L cells expressing naive B1-8-IgM-BCR (**A**) and primary naive B cells expressing B1-8-IgM-BCR (**B**). Bars represent mean ±SEM. Two-tailed *t* tests were performed for the statistical comparisons. Data are from at least 40 cells over three independent experiments. (**C**) Representative TIRFM images showing the dynamics of the synaptic accumulation of IgM-BCRs from J558L cells expressing naive B1-8-IgM-BCR in contact with coverslip presenting 12 pN, 43 pN or 56 pN NP-TGT sensors at the indicated time points. Scale bar is 1.5 μm. (**D**, **E**) Comparisons of averaged traces showing the dynamic accumulation of naive IgM-BCRs into the immunological synapse (**D**) and the growing features of the size of contact area (**E**) for J558L cells expressing naive B1-8-IgM-BCR as demonstrated in (**C**) in a 10 min TIRFM imaging time course. Bars represent mean ±SEM. Two-tailed *t* tests were performed for the statistical comparisons. Data are from at least 20 cells over two independent experiments.

*Figure 2. continued on next page*

*Figure 2. Continued*

The following figure supplements are available for figure 2:

**Figure supplement 1**. The contact area after IgM-BCR activation is dependent on mechanical forces with multi-threshold effects and such a pattern is still evident at low density of NP-TGT sensor.
**Figure supplement 2**. The patterned dependence on the mechanical forces of IgM-BCR activation does not rely on BCR internalization.

---

*Natkanski et al., 2013*). These results suggest that the internalization of BCR and antigen molecules did not contribute to the different levels of the accumulation of the IgM-BCRs into the B cell IS in response to different NP-TGT sensors with different mean rupture forces. Indeed, B cells that were pretreated with monodansylcadaverine (MDC), an inhibitor to block B cell internalization (*Tolar et al., 2005*; *Liu et al., 2010a*), also exhibited the dependent on mechanical forces in the synaptic accumulation of the IgM-BCRs (*Figure 2—figure supplement 2D–F*).

## The volume of the IgM-BCR microcluster produced by different NP-TGT sensors is dependent on mechanical forces and exhibits a similar multi-threshold effect

BCR microclusters have been demonstrated to serve as the most basic platform for the investigation of the initiation of BCR signaling by our studies and those of others (*Harwood and Batista, 2010*; *Pierce and Liu, 2010*). We thus assessed the volume of the BCR microclusters produced by the series of NP-TGT sensors by quantifying the fluorescence intensity (FI) of each individual BCR microcluster. We mathematically fitted each BCR microcluster using a 2D Gaussian function as reported earlier (*Source code 1*) (*Liu et al., 2010a*). We similarly observed that the volume of the BCR microclusters is dependent on the strength of the mechanical force with 56 pN NP-TGT sensors producing much bigger and brighter BCR microclusters than 43 pN and 12 pN NP-TGT molecules (*Figure 3A,B*). The multi-threshold effect was also observed when we analyzed the volume of the generated BCR microclusters. NP-TGT molecules with low mean rupture force (12–16 pN) triggered the formation of small BCR microclusters, whereas NP-TGT sensors with medium mean rupture force (23–43 pN) initiated medium-level volume BCR microclusters, and NP-TGT molecules with high mean rupture force (50–56 pN) accounted for bigger and brighter BCR microclusters (*Figure 3A,B*). Thus, the volume of the IgM-BCR microcluster produced by different NP-TGT sensors is dependent on mechanical forces and exhibits a similar multi-threshold effect.

## The strength of IgM-BCR signaling is dependent on mechanical forces

Next, we examined the initiation of BCR signaling in B cells when encountering the 12 pN, 43 pN, and 56 pN NP-TGT molecules by quantifying the accumulation of pSyk, pPLCγ2, and pTyr molecules into the B cell IS using the TIRFM image analysis method as reported in our earlier studies (*Liu et al., 2010b*) and those of others (*Fleire et al., 2006*; *Liu et al., 2012a*). All of these signaling molecules play essential roles in the transmembrane signaling transduction of BCR. For example, Syk binds to the phosphorylated Immunoreceptor tyrosine-based activation motif (ITAM) on the cytoplasmic tail of Igα and Igβ, and subsequently Syk will undergo auto-phosphorylation at multiple tyrosine sites within its linker regions to be converted to a signaling active form (*Saouaf et al., 1994*). The phosphorylated Syk also provides docking sites for PLCγ2 (*Weber et al., 2008*). We found that the membrane proximal recruitment of each of these signaling

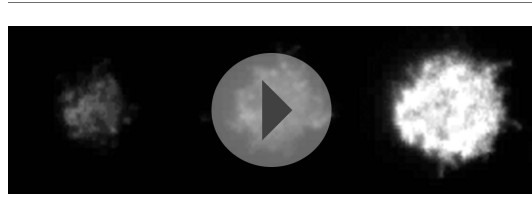

**Video 2.** Representative time lapse TIRFM images showing the dynamics of the synaptic accumulation of IgM-BCRs from J558L cells expressing naïve B1-8-IgM-BCR in contact with coverslip presenting 12 pN, 43 pN, or 56 pN NP-TGT sensor. Scale bar is 1.5 µm. The video was recorded with a 4-s time interval and is shown at 30 frames per second. Related to *Figure 2C*.

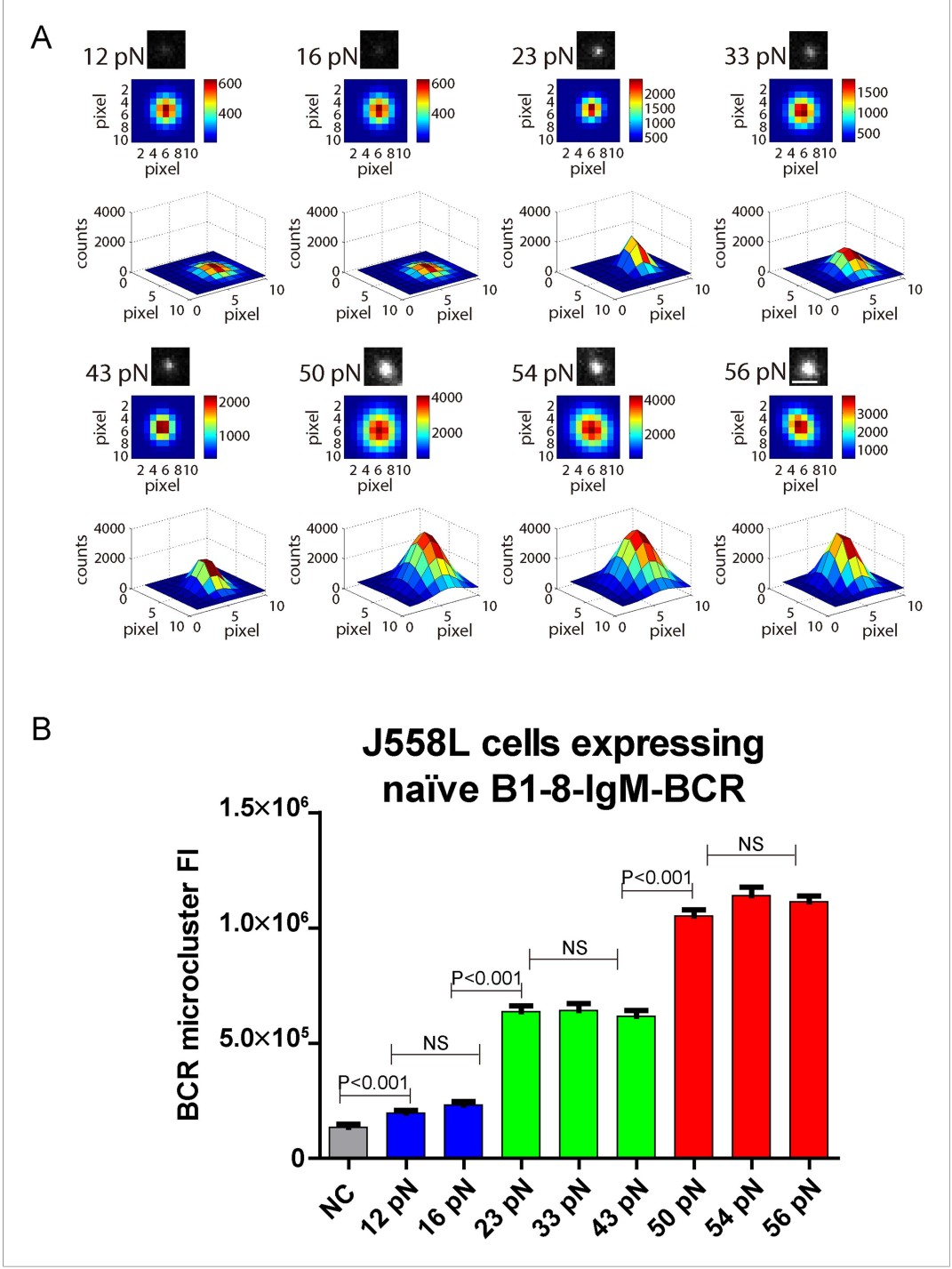

Figure 3. The volume of the IgM-BCR microcluster produced by different NP-TGT sensors is dependent on mechanical forces and exhibits a similar multi-threshold effect. (**A**) Representative original (top panel), pseudo-colored 2D (middle panel), and 2.5D Gaussian images (bottom panel) of typical BCR microclusters induced by 12, 16, 23, 33, 43, 50, 54, and 56 pN NP-TGT sensors. Scale bar is 1.5 μm. (**B**) Statistical comparison of FI of hundreds of BCR microclusters in the immunological synapse in J558L cells expressing naïve B1-8-IgM-BCR encountering NP-TGT sensors with indicated tension force. Bars represent mean ±SEM. Two-tailed *t* tests were performed for the statistical comparisons. Data are from at least 30 cells over three independent experiments.

molecules is dependent on mechanical forces with the 56 pN NP-TGT sensors showing the highest recruitment, while the 43 pN NP-TGT sensors had a medium level of recruitment and the 12 pN NP-TGT sensors had the lowest level of recruitment (*Figure 4A–C*), consistent with the results from the above mentioned experiments quantifying the synaptic accumulation of the BCR molecules (*Figure 2A,B*). We also analyzed the volume of these signaling molecule microclusters by quantifying the FI of the microclusters and confirmed that the FI of pSyk, pPLCγ2, and pTyr microclusters produced by different NP-TGT molecules is dependent on mechanical force with a similar multi-threshold effect (*Figure 4D–F*). Thus, it was clear that the strength of the initiated IgM-BCR signaling is dependent on mechanical forces.

## The outside-in activation of integrin enhances the strength of IgM-BCR activation but does not change its patterned dependence on mechanical forces

We explored whether the conventional mechanosensor integrin LFA-1 expressed on the surface of B cells influences the patterned dependence on the mechanical forces in B cell activation. To address this question, we co-tethered both NP-TGT sensors and the adhesion molecule, intercellular adhesion molecule 1 (ICAM-1, which is a ligand for integrin LFA-1) on the surface of coverslip and similarly performed TIRFM imaging experiments. The addition of ICAM-1 dramatically enhanced the synaptic accumulation of the BCRs (*Figure 5A*), suggesting the increased sensitivity to antigen stimulation in the initiation of B cell activation, consistent with the published studies in both the BCR (*Carrasco et al., 2004*) and TCR system (*Bachmann et al., 1997*; *Dustin, 2009*). However, it was clear that the outside-in activation of integrin did not change the fact that B cell activation is dependent on mechanical forces with a similar multi-threshold effect (*Figure 5B*). To further confirm this conclusion, we inactivated the function of focal adhesion kinase (FAK), a member of the

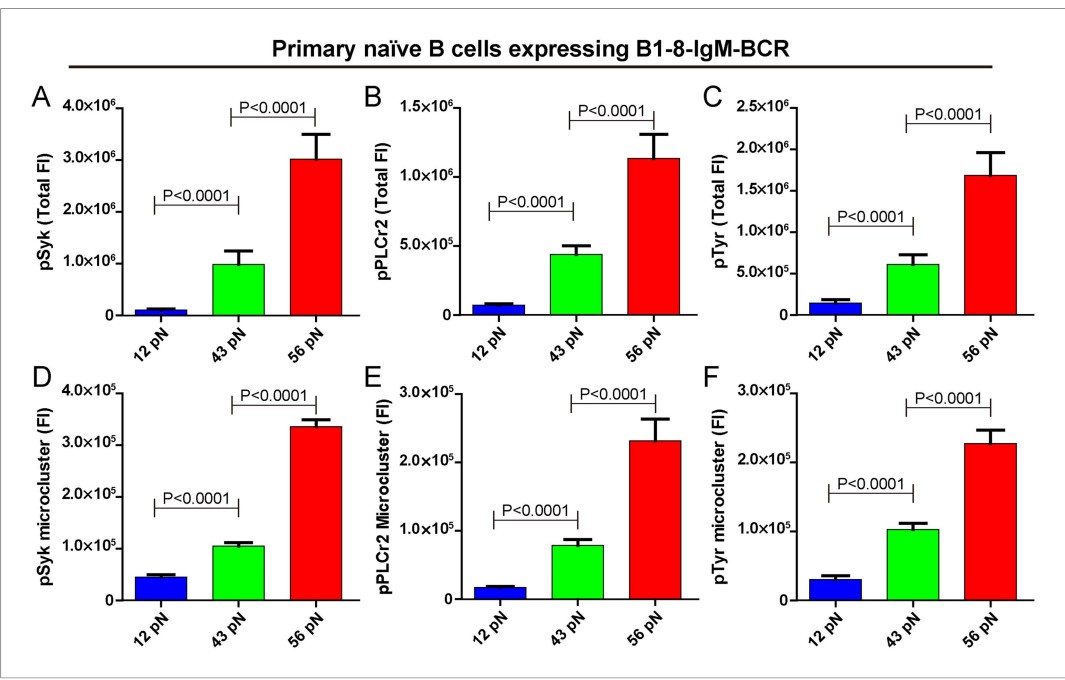

**Figure 4**. The strength of IgM-BCR signaling is dependent on mechanical forces. (**A–C**) Statistical quantification of the synaptic recruitment of pSyk (**A**), pPLCγ2 (**B**), and pTyr (**C**) in primary naive B cells expressing B1-8-IgM-BCR that were placed on coverslip presenting 12 pN, 43 pN or 56 pN NP-TGT sensors. (**D–F**) Statistical comparison of the volume of pSyk (**D**), pPLCγ2 (**E**), or pTyr (**F**) microclusters in J558L cells expressing naive B1-8-IgM-BCR that were produced by 12 pN, 43 pN, or 56 pN NP-TGT molecules. In all of these plots, bars represent mean ±SEM. Two-tailed *t* tests were performed for the statistical comparisons. Data were at least from 30 cells of three independent experiments.

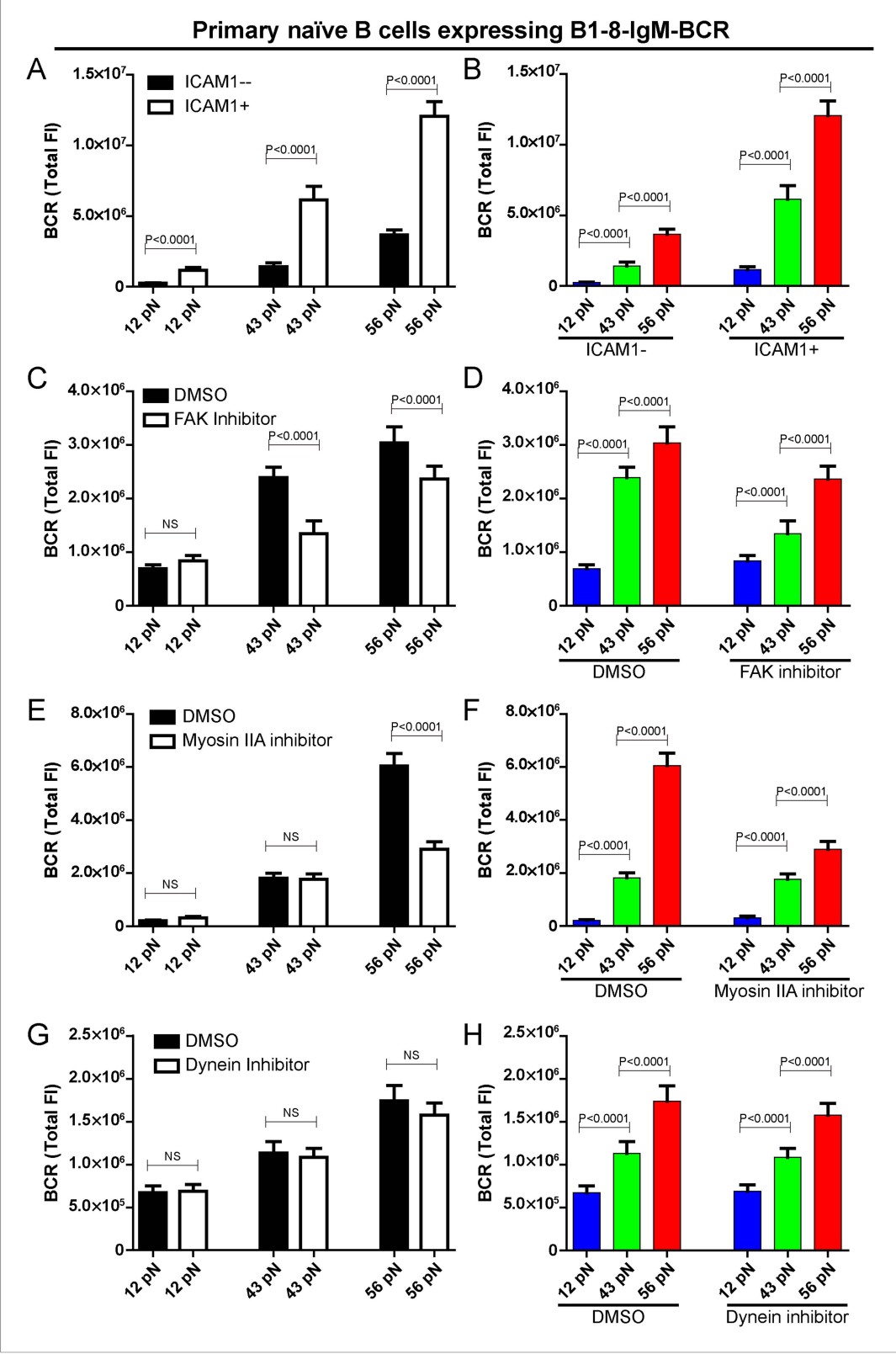

**Figure 5**. The patterned dependence on the mechanical forces of IgM-BCR activation does not rely on LFA-1 mediated adhesion and dynein, and is only partially dependent on myosin IIA. (**A**, **B**) The synaptic accumulation of IgM-BCRs in primary naive B cells expressing B1-8-IgM-BCR in contact with the indicated types of NP-TGT sensors
*Figure 5. continued on next page*

*Figure 5. Continued*

with or without ICAM-1 co-tethering. Cross comparison strategy were used in these figures. (**C**–**H**) The synaptic accumulation of IgM-BCRs in primary naive B cells expressing B1-8-IgM-BCR in contact with the indicated types of NP-TGT probes. In this experiment, primary naive B cells expressing B1-8-IgM-BCR were pretreated with DMSO as a control in combination with FAK inhibitor (**C**, **D**), myosin IIA inhibitor (**E**, **F**) or dynein inhibitor (**G**, **H**) following a protocol that was detailed in 'Materials and methods' section. Cross comparison strategy were used in these figures. In all of these plots, bars represent mean ±SEM. Two-tailed *t* tests were performed for the statistical comparisons. Data were at least from 30 cells in each group of three independent experiments.

The following figure supplement is available for figure 5:

**Figure supplement 1**. Functional test of pharmaceutical inhibitors.

non-receptor protein-tyrosine kinase family, that is known to play a key role in the activation of integrin signaling pathways (*Slack-Davis et al., 2007*; *Yu et al., 2012*; *Bashour et al., 2014*). We pretreated B cells with FAK-specific inhibitor PF573-228 (*Slack-Davis et al., 2007*). We found that the pretreated B cells still maintained the general patterned dependence on mechanical forces with a multi-threshold effect (*Figure 5C,D*, *Figure 5—figure supplement 1A*). However, it was clear that the high-end (56 pN) and medium-level (43 pN) but not low-end (12 pN) mechanical force threshold was more influenced by the inactivation of integrin, suggesting that the breakthroughs of the medium-level and high-end threshold of mechanical forces are partially supported by the inside-out activation of integrin.

## The high-end but not low-end mechanical force threshold is dependent on myosin IIA in IgM-BCR activation

Motor proteins including myosin IIA and dynein are known to play important roles in B cell activation. There are recent studies showing that B cells utilize the contraction forces generated by myosin IIA to rupture the interaction between the BCR and antigen molecules (*Natkanski et al., 2013*), while dynein is required for the retrograde motile feature of the BCR microclusters into the center of B cell IS (*Schnyder et al., 2011*). Thus, we assessed the contribution of these two motor proteins to the patterned dependence on mechanical forces during IgM-BCR activation. Unexpectedly, the inactivation of myosin IIA did not dramatically change the dependence on mechanical forces, however we observed a significant drop in the strength of the synaptic accumulation of the BCRs for the high threshold NP-TGT sensor at 56 pN only, and no drop was observed for the medium-level threshold NP-TGT sensor at 43 pN, nor the low threshold NP-TGT sensor at 12 pN (*Figure 5E,F*, *Figure 5—figure supplement 1B*). On the other hand, when inactivating dynein, we did not observe significant changes in the BCR accumulations compared to DMSO controls (*Figure 5G,H*, *Figure 5—figure supplement 1C*). All of these results suggest that the breakthroughs of the high-end but not the medium-level and low-end thresholds of mechanical forces are supported by myosin IIA.

## The activation of isotype-switched IgG-BCRs or IgE-BCRs on memory B cells requires either no tension or a mechanical force below 12 pN

B cells use different isotypes of BCRs to recognize antigens and to initiate transmembrane activation signaling. Mature naive B cells use IgM-BCRs (termed naive IgM-BCR thereafter), while memory B cells mainly use IgG-BCRs (termed memory IgG-BCR thereafter) with a fraction of these use IgE-BCRs (termed memory IgE-BCR thereafter). Memory B cells are responsible for the rapid antigen recall humoral responses upon vaccine immunization (*McHeyzer-Williams and McHeyzer-Williams, 2005*; *Pierce and Liu, 2010*). Here, we explore the sensitivities and thresholds toward mechanical forces in the activation of memory B cells expressing isotype-switched IgG-BCRs or IgE-BCRs. We addressed this question using J558L cells expressing memory B1-8-IgG-BCR in the same experimental system that has been used for the naive B1-8-IgM-BCR-expressing J558L cells as described in detail above. Surprisingly, we observed a totally different pattern in terms of the dependence on mechanical forces for the activation of memory IgG-BCRs. Each of the three NP-TGT sensors producing mechanical forces at 12, 43, or 56 pN similarly drove the cells to undergo spreading responses and the synaptic

accumulation of BCRs or pSyk signaling molecules into the IS (*Figure 6A–D*). This suggested that the activation of memory IgG-BCRs requires either no tension or a mechanical force below the mean rupture force of the lowest-force NP-TGT we used (12 pN). Similar patterned activation of memory IgG-BCR was observed at different surface density of NP-TGT sensors on coverslip (*Figure 6—figure supplement 1A–C*). Furthermore, a similar phenomenon was observed when examining the activation of B1-8 primary B cells expressing memory IgG-BCRs that were derived from an in vitro class-switch response following our published protocol (*Liu et al., 2010b*) (*Figure 6E,F*). We also used high speed TIRFM imaging to test the dynamics of memory IgG-BCR accumulation into the B cell IS starting with

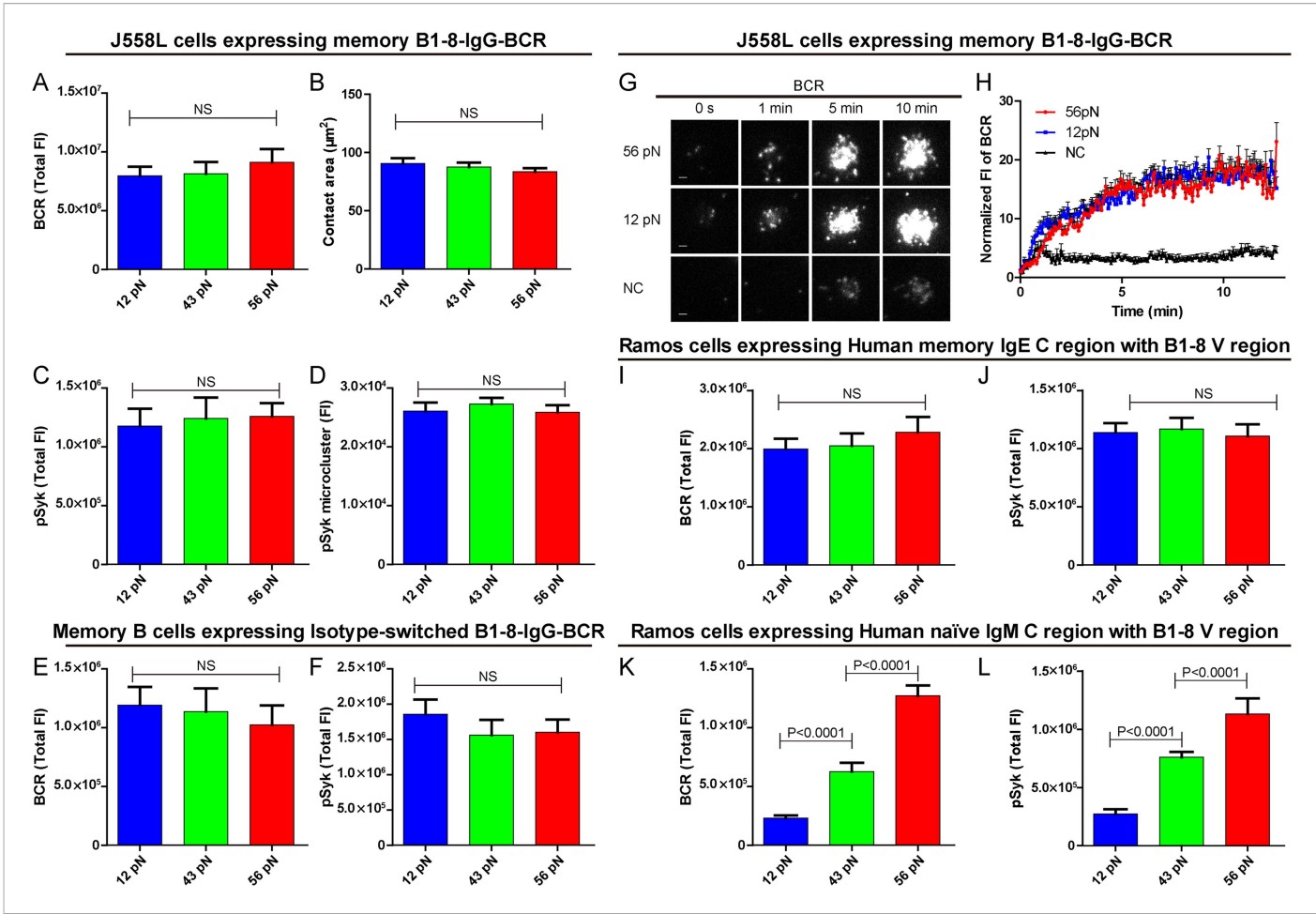

**Figure 6**. The activation of isotype-switched IgG-BCRs or IgE-BCRs on memory B cells requires either no tension or a mechanical force below 12 pN. (**A**, **B**) Statistical quantification of the synaptic accumulation of IgG-BCR and the volume of the contact area of J558L cells expressing memory B1-8-IgG-BCR encountering 12 pN, 43 pN, or 56 pN NP-TGT sensors. (**C**, **D**) Statistical analyses of synaptic accumulation of pSyk accumulation (**C**) and the volume of pSyk microcluster (**D**) in response to 12 pN, 43 pN, or 56 pN NP-TGT sensors. (**E**, **F**) Quantification of the synaptic accumulation of IgG-BCRs (**E**) or pSyk (**F**) in memory B cells expressing isotype-switched B1-8-IgG-BCR from B1-8 Tg mice that were placed on coverslip presenting 12 pN, 43 pN, or 56 pN NP-TGT probes. (**G**) Representative TIRFM images showing the dynamics of the synaptic accumulation of IgG-BCRs from J558L cells expressing memory B1-8-IgG-BCR in contact with coverslip presenting 12 pN, 56 pN NP-TGT sensor, or control TGT (NC) molecule at the indicated time points. Scale bar is 1.5 μm. (**H**) Comparisons of averaged traces showing the dynamic accumulation of memory IgG-BCRs as demonstrated in (**G**) in a 13 min TIRFM imaging time course. Bars represent mean ±SEM. Data were from at least 20 cells over three independent experiments. (**I–L**) Statistical analyses of the synaptic accumulation of two types of chimeric BCRs and pSyk, Human mIgE heavy chain (**I**, **J**), or Human mIgM heavy chain (**K**, **L**) with mouse B1-8 variable region in human Ramos B cells encountering 12 pN, 43 pN, or 56 pN NP-TGT sensors. In all of these plots, bars represent mean ±SEM. Two-tailed *t* tests were performed for the statistical comparisons. Data were from at least 30 cells in each group of three independent experiments.

The following figure supplement is available for figure 6:

**Figure supplement 1**. Quantification of the accumulation of IgG-BCR in recognition of NP-TGT sensors at different surface density.

the initial immune recognition of memory IgG-BCRs with 12 pN, 56 pN and NP non-conjugated control TGT molecules within a 13-min time course (*Figure 6G,H*, *Video 3*). It was clear that 12 pN and 56 pN NP-TGT molecules similarly drove the accumulation of BCRs into the B cell IS. All of these results show that the activation of the memory IgG-BCR is independent on mechanical forces ranging from 12 to 56 pN as tested in our NP-TGT experimental system.

Next, we examined the dependence on mechanical forces in terms of the activation of memory IgE-BCRs in Ramos human B cells expressing memory B1-8-IgE-BCRs. Here, we similarly observed that the activation of memory IgE-BCRs require either no tension or a mechanical force below 12 pN (*Figure 6I,J*). As a system control, it was clear that the activation of the naive B1-8-IgM-BCRs expressed in Ramos B cells was also dependent on mechanical force with a multi-threshold effect (*Figure 6K,L*). Conclusively, it was clear that the activation of the memory IgG-BCRs or IgE-BCRs requires either no mechanical force or a force lower than that provided by the 12 pN NP-TGT, highlighting the significant low threshold for the activation of memory IgG-BCR or IgE-BCR that are typically expressed on memory B cells compared to the case of the naive IgM-BCR on mature naive B cells. All these results provide new explanations for the rapid and high-titered IgG antibody responses upon re-encounter with antigen by memory B cells.

## The lower mechanical force threshold of memory IgG-BCR activation is dependent on its cytoplasmic tail

The unexpected extremely low mechanical force threshold (might be less than 12 pN or even lower) for the memory IgG-BCR but not the naive IgM-BCR activation drove us to ask the next question: why does the memory IgG-BCR behave differently from the naive IgM-BCR? This question is especially intriguing as both memory IgG-BCR and naive IgM-BCR use the exact same BCR component, the Igα and Igβ heterodimer, to initiate BCR signaling. In contrast, the mIgG and the mIgM are the BCR components to recognize the antigens (*Schamel and Reth, 2000*; *Tolar et al., 2005*). In addition to the constant region of the Ig, mIgG and mIgM also differ greatly in their cytoplasmic domain. In fact, mIgM has only three-amino acids in its cytoplasmic tail (KVK), while all mIgG subtypes have 28-amino acid cytoplasmic tails, which are extremely conserved across species (*Reth, 1989*; *Tarlinton, 1997*; *Liu et al., 2010a*, *2012b*). Early mice model studies utilizing biochemical assays and live cell imaging demonstrated that the cytoplasmic tail of mIgG is both necessary and sufficient to confer an enhanced activation of IgG-BCR expressing memory B cells compared to the case of IgM-BCR expressing naive B cells (*Kaisho et al., 1997*; *Martin and Goodnow, 2002*; *Wakabayashi et al., 2002*; *Horikawa et al., 2007*; *Waisman et al., 2007*; *Engels et al., 2009*; *Liu et al., 2010b*, *2012b*; *Engels et al., 2014*). To examine the contribution of the cytoplasmic tail of mIgG in the low mechanical force threshold for the activation of the memory IgG-BCR, we took advantage of the four types of J558L cells expressing naive B1-8-IgM-BCR and memory B1-8-IgG-BCR with cytoplasmic tail swapped forms as reported in our previous study (*Liu et al., 2010b*): (1) memory B1-8-IgG-BCR; (2) mIgG swapped with a mIgM cytoplasmic tail (memory B1-8-IgG-BCR equipped with a mIgM cytoplasmic tail, termed GGM thereafter); (3) naive B1-8-IgM-BCR; (4) mIgM swapped with a mIgG cytoplasmic tail (naive B1-8-IgM-BCR equipped with a mIgG cytoplasmic tail, termed MMG thereafter) (*Figure 7A*). We found that the J558L cells expressing GGM showed the force dependent activation by accumulating more BCRs and pSyk into the IS with the higher mean rupture force NP-TGT molecules, similar to the case of J558L cells expressing naive B1-8-IgM-BCR (*Figure 7B–D*). In contrast, it was apparent that the J558L cells expressing MMG behaved similarly to the case of J558L cells expressing memory B1-8-IgG-BCR in a force-independent manner (*Figure 7E–G*). All of these results suggested that the lower mechanical force threshold of memory IgG-BCR activation depends on its cytoplasmic tail.

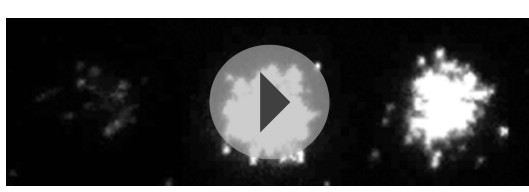

**Video 3.** Time lapse images showing the dynamics of the synaptic accumulation of IgG-BCRs from J558L cells expressing memory B1-8-IgG-BCR in contact with coverslip presenting 12 pN, 56 pN NP-TGT or control TGT (NC) sensor at the indicated time points. Scale bar is 1.5 μm. The video was recorded with a 4-s time interval and is shown at 30 frames per second. Related to *Figure 6G*.

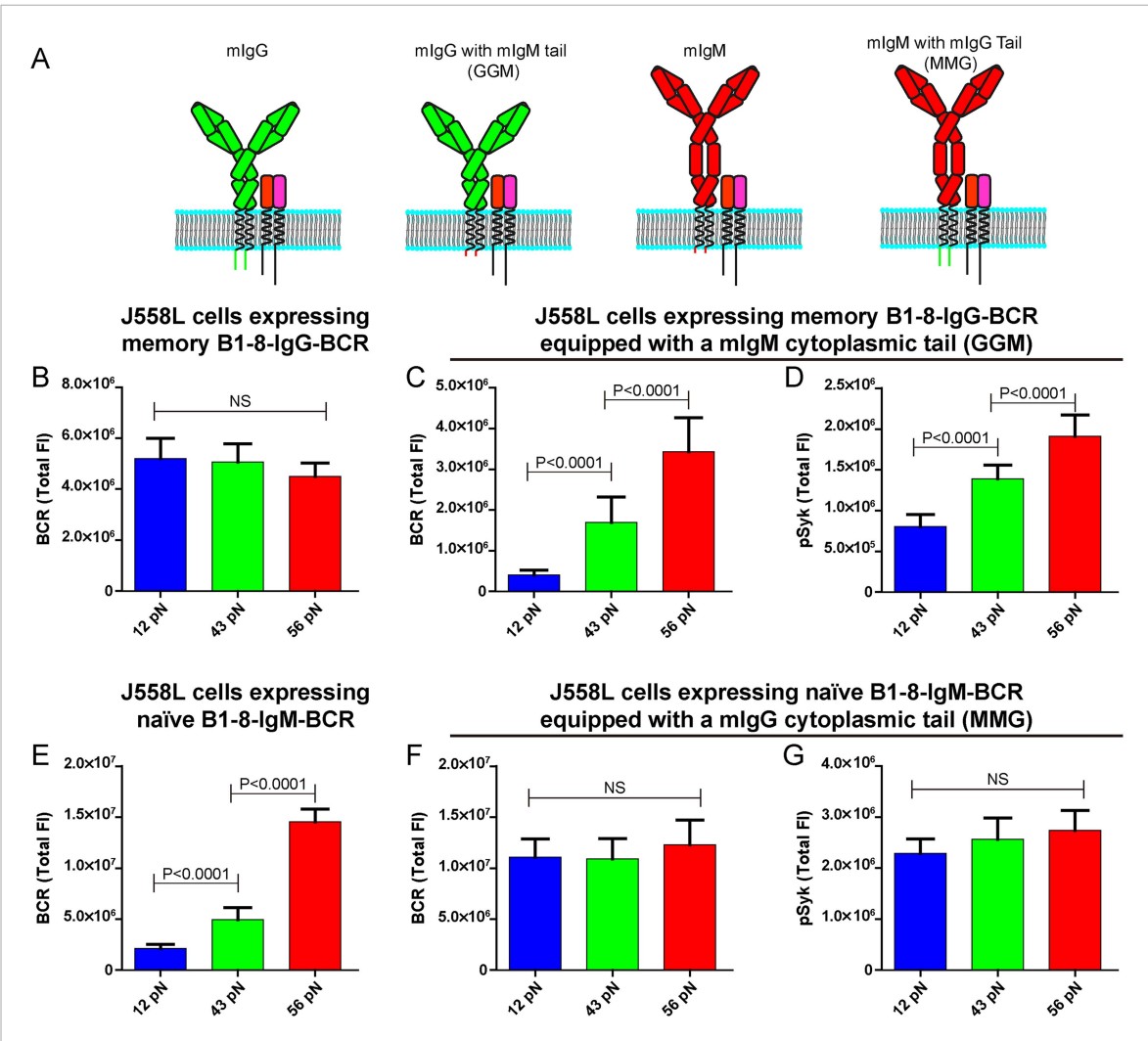

**Figure 7**. The lower mechanical force threshold of IgG-BCR activation is dependent on its cytoplasmic tail. (**A**) Schematic illustration of the strategy of swapping the cytoplasmic tail of B1-8-IgG- or B1-8-IgM-BCRs. (**B–G**) Quantification of the synaptic accumulation of BCRs in J558L cells expressing memory B1-8-IgG-BCR (**B**), J558L cells expressing memory B1-8-IgG-BCR equipped with a mIgM cytoplasmic tail, termed GGM (**C**), J558L cells expressing naïve B1-8-IgM-BCR (**E**) and J558L cells expressing naïve B1-8-IgM-BCR equipped with a mIgG cytoplasmic tail, termed MMG (**F**). Also given are the synaptic accumulations of pSyk in GGM (**D**) and MMG cells (**G**). In all of these plots, bars represent mean ±SEM. Two-tailed *t* tests were performed for the statistical comparisons. Data were from at least 30 cells in each group of three independent experiments.

## Discussion

B cells produce protective antibodies against pathogens. Antibody response is originated from the activation of B cells upon BCR and antigen recognition. As mentioned earlier, the antigens recognized by B cells in vivo exhibit great diversities including antigen density (*Fleire et al., 2006*; *Liu et al., 2010a*), antigen affinity (*Fleire et al., 2006*; *Liu et al., 2010a*), antigen valency (*Bachmann et al., 1993*; *Liu et al., 2004*; *Liu and Chen, 2005*), Brownian mobility feature of the antigen (*Wan and Liu, 2012*), and the stiffness of the substrates presenting the antigen (*Wan et al., 2013*; *Zeng et al., 2015*). All these studies demonstrated that BCR is an extraordinary receptor which can efficiently discriminate the chemical and physical features of an antigen ligand. That raises a long-standing question in immunological studies that how possible the strength of BCR activation can be so complicated and efficient to combine sensitivity, threshold, speed, specificity, affinity, and substrate stiffness discriminatory power?

In this report, we modified the dsDNA-based TGT developed by Ha and his colleagues (*Wang and Ha, 2013*) to acquire a series of 8 NP-TGT sensors with predefined mean rupture force from low to high, respectively (12, 16, 23, 33, 43, 50, 54 and 56 pN at the condition in the original study). The use of these NP-TGT molecules in combination with the high resolution high speed live cell imaging technique through TIRFM allows us to investigate several intriguing questions: (1) Does the BCR itself possess any mechanosensing properties? (2) Is there a threshold for BCR activation from the mechanical forces that are delivered to BCR by antigen? (3) How sensitive the BCR will be toward different mechanical forces above that threshold in BCR activation?

Supported by NP-TGT system, for the first time, we demonstrated that IgM-BCR activation is in a mechanical force dependent manner by providing evidence that high mean rupture force NP-TGT sensors which can stand for high-force load-triggered more aggressive spreading response and the enhanced BCR signaling than low mean rupture force NP-TGT sensors. Since such a patterned dependence on mechanical force in the initiation of IgM-BCR does not strictly rely on integrin or adhesion molecule, we propose that IgM-BCR alone could function as a mechanosensor. The performance of IgM-BCRs in TGT-based mechanical force experimental system is mostly consistent with the reported findings on integrin molecules (*Wang and Ha, 2013*). However IgM-BCR and integrin also exhibit some obvious differences in their respective responding pattern to the same mechanical force spectrum (12 pN–56 pN) produced by TGT sensors.

First of all, 12 pN NP-TGT sensor, the one enduring the smallest mechanical force, can induce the mild but consistent activation of B1-8-naive IgM-BCR when was compared to the control TGT without the conjugated NP antigen. Though, it is a fact that the strength of such activation produced by 12 pN NP-TGT sensor is dramatically weaker than those produced by other NP-TGT molecules bearing higher force (such as 56 pN NP-TGT). In contrast, integrin does not seem to get activated under 12 pN or even 16 pN TGTs (*Wang and Ha, 2013*). These results suggested that the initiation of BCR signaling could be extremely sensitive to mechanical forces. BCR might rely on such fine sensitivity to initiate its signaling cascade during the surveillance for non-self-antigens that could be of very low density. Subsequently, those initially weak signals could be further boosted by other well-characterized mechanisms through activating co-receptors (such as CD19) (*Depoil et al., 2008*) or co-stimulatory factors (such as LFA-1) (*Spaargaren et al., 2003*; *Carrasco et al., 2004*; *McLeod et al., 2004*; *Dustin et al., 2006*; *Tseng et al., 2008*; *Arana et al., 2008a*, *2008b*).

Secondly, IgM-BCR exhibits an obvious multi-threshold effect in response to the NP-TGT sensors enduring different mechanical forces from 12 pN to 56 pN. The low-force 12–16 pN NP-TGT sensors triggered a weak activation, whereas the middle-force 23–43 pN NP-TGT sensors initiated a medium-level activation, and the high-force 50–56 pN NP-TGT sensors accounted for a strong activation. Within these three thresholds, there is a quite obvious barrier effect as an increase in mechanical force, such as from 23 pN to 33 pN or to 43 pN within the medium-level mechanical force barrier, will not result in any dramatically enhanced activation responses of IgM-BCRs. Instead, the activation of IgM-BCR is extremely sensitive to the changes of mechanical forces exceeding these barriers, such as the increase of the mechanical force from 12 pN to 43 pN or from 43 pN to 56 pN. We speculate that these observations might reflect some key features of dsDNA-derived mechanical force sensor that each sensor represents a distribution of force value with an FWHM of 5 pN for unzipping rupture mode and 15 pN for shearing rupture mode (*Lang et al., 2004*). Thus, the actual rupture force of any given TGT sensor shall not be viewed as a fixed value, but rather a distribution with 'shoulder overlap' to the nearby TGT sensors. We also speculate that these findings would potentially deliver key meanings with biological significance. Different from many other receptors, IgM-BCR composes a mIgM and a heterodimer of Igα and Igβ in a 1 mIg: 1 Igα–Igβ stoichiometry with mIgM, the biosensor for antigens, and Igα–Igβ heterodimer, the BCR signaling initiator (*Schamel and Reth, 2000*; *Tolar et al., 2005*). The antigenic signal needs to be delivered from the antigen binding region on the extracellular domain of mIgM to the signaling active ITAM on the cytoplasmic domain of both Igα and Igβ. It is still a big open question in terms of the molecular mechanism of how exactly BCR transduces physical and chemical signals. Here, we tried to interpret our data in a way that such multi-threshold effect of the activation of IgM-BCRs reflects the multi-threshold progress of delivering these mechanical forces induced potential conformational changes from mIgM to Igα and Igβ heterodimer complex. It is also our speculation that such multi-threshold likely provides a chance for the proof-testing of the mechanical forces applied to IgM-BCR during its activation. All these speculations are under our further extensive investigation.

These observed differences that were induced by different NP-TGT sensors cannot exclude the contribution of lifetime of the bonds between BCR and NP-TGT sensors. Although, the contribution from mechanical forces and the lifetime of the BCR with the NP-TGT molecules cannot be absolutely separated. This especially intrigues a question when considering the recent study by Zhu and his colleagues (*Liu et al., 2014*), showing that the mechanical force prolongs lifetime of catch bond between TCR and agonist pMHC, but decreases the lifetime of slip bond between TCR and antagonist pMHC antigen. The authors showed that both the value of the force and the lifetime of the force are important factors in TCR activation. The situation with BCR shall be more complicated as different from TCR, BCR is a bivalent binding receptor, stating that one BCR can bind to two NP-TGT molecules. Additionally, the recent advance of the super-resolution imaging studies demonstrated that BCRs form obvious nanoclusters with 20–50 IgM-BCRs per nanocluster on quiescent B cells (*Mattila et al., 2013*). All these features of BCR demonstrated the high possibility of multivalent bonds based mechanical forces in regulating the initiation of B cell activation. Additionally, Tolar and his colleagues recently used AFM to examine the rupture forces between B1-8-IgM-BCR and NP antigen (*Natkanski et al., 2013*). Their study provides useful reference information to better understand the relationship between rupture forces and lifetime of a single bond B1-8-IgM-BCR and NP antigen.

It shall be noted that the values of the rupture forces of TGT sensors were all calculated based on the data measured by magnetic tweezers system with a loading rate of 0 pN/s (constant force with a time under force mode of 2 s during the measurements). Rupture force measuring systems using AFM-based vs magnetic tweezers-based approaches show many differences including but not limited to different spring constants, different loading rate and different time under load (*Albrecht et al., 2003*; *Lang et al., 2004*; *Hatch et al., 2008*). Thus, all these differences prevented the feasibility of a direct comparison of the mechanical force value that is acquired from different kind of measurements (constant force vs constant loading). From this point of view, it would be of great importance to use the constant force measurements to examine the interaction of B1-8-BCR with NP antigen, and to investigate how the magnitude and duration of mechanical forces will influence B cell activation. During the measurement, the time of the force under load shall be carefully picked up because it is known that such a parameter greatly influence the measured value of the rupture forces (*Lang et al., 2004*; *Natkanski et al., 2013*; *Wang and Ha, 2013*). As mentioned above a constant force mode with a time under force of 2 s was used to measure the rupture force of these TGT sensors by magnetic tweezers (*Wang and Ha, 2013*). The relevant timescale of rupturing NP-TGT sensors in our experimental system shall be at the seconds to dozens of seconds level with consideration from the recent studies by Tolar and his colleagues (*Natkanski et al., 2013*).

In this report, we examined the molecule requirements of the patterned dependence on mechanical forces in IgM-BCR activation. The source accounting for the applied force would be quite diverse in a live cell. In the case of BCR, motor proteins including myosin IIA and dynein molecules are known to be the mechanical force provider. Myosin IIA executes the vertical forces to BCR microclusters to rupture the bonds between BCRs and membrane-bound antigens, and such event is used by B cells to discriminate antigen affinity (*Natkanski et al., 2013*). Dynein is shown to be required for the retrograde motile feature of BCR microclusters into the center of the B cell IS, suggesting dynein would more likely execute a lateral force (*Schnyder et al., 2011*). We assessed the contribution of myosin IIA and dynein in the patterned dependence on mechanical forces of IgM-BCR activation. Unexpectedly, the inactivation of either myosin IIA or dynein did not dramatically changed the patterned dependence on mechanical forces in IgM-BCR activation, suggesting the formation of the basic barrier for the patterned multi-threshold effect does not strictly reply on myosin IIA or dynein. However, in primary naive B cells expressing B1-8-IgM-BCR expressing B cells with inactivated myosin IIA, we observed a significantly decreased accumulation of BCR in response to high-end 56 pN NP-TGT sensor only, and such drop was not observed in the case of medium-level threshold 43 pN NP-TGT sensor or low-end 12 pN NP-TGT sensor. No such drop was observed in each of these multi-threshold NP-TGT molecules after inactivating dynein. All these data drive us to speculate that the breakthrough of the high-end but not the medium-level or low-end mechanical force threshold might be contributed by myosin IIA. In contrast, dynein does not seem to play an obvious role in the establishment of the barriers of mechanical forces.

The unique multi-threshold pattern in IgM-BCR activation is not strictly mediated by the outside-in or inside-out activation of the integrin molecules as addition of neither ICAM-1 nor FAK inhibitor

changed that patterned dependence on mechanical forces in IgM-BCR activation. However, in FAK inactivated B cells, we observed a significantly decreased accumulation of BCR in response to high-end 56 pN NP-TGT sensor and medium-level threshold 43 pN NP-TGT sensor, while no such drop was observed in the case of low-end 12 pN NP-TGT sensor. We speculate from these data that the breakthrough of the high-end and medium-level threshold but not low-end mechanical force barriers might be contributed by the inside-out activation of integrin. This is in consistent with the role of myosin IIA as discussed above. Myosin IIA bound actin filaments are physically connected with integrin through the key focal adhesion molecules talin and vinculin and was shown to be important for the function of TCR microclusters (*Ilani et al., 2009*). Inactivation of either myosin IIA or FAK affects the high-end mechanical force barrier for IgM-BCR activation seems to suggest that these two molecules may regulate B cell activation through similar mechanism at least partially.

The most striking observation of this report is that the activation of isotype-switched IgG-BCR and IgE-BCR on memory B cells only requires either no tension or a mechanical force below 12 pN since 12 pN, 43 pN or 56 pN NP-TGT sensors similarly drove the activation of IgG-BCR (or IgE-BCR). All these results highlighted the significant low-force requirement to initiate the activation of IgG-BCR or IgE-BCR that are expressed on memory B cells compared to the case of IgM-BCR on mature naive B cells, implicating a new possible mechanism to explain the rapid and high-titered IgG antibody responses upon re-encounter with antigen of memory B cells. It is worth noting that the recent study by Ha and his colleagues showed that the activation of Notch receptor also requires either no tension or a single molecule force smaller than 12 pN (*Wang and Ha, 2013*). It is of great interest to address the underlying fundamental mechanism of the different mechanical force requirements for different receptors. In this report, we investigated which domain of mIgG resulted in the low or none requirements on mechanical forces for IgG-BCRs compared to the case of IgM-BCRs since both of these two forms of BCRs use the same signaling initiation component, Igα and Igβ heterodimer. By a systemic swap experiment we demonstrated that the lower threshold on mechanical forces of memory IgG-BCR activation is dependent on its cytoplasmic tail. At this moment, we are addressing the question of why an evolutionarily conserved cytoplasmic tail confers such low threshold on mechanical forces for the activation of memory IgG-BCR. We hypothesize that the cytoplasmic tail of mIgG (or mIgE) might have some capabilities to facilitate the conformational changes of the 'umbrella-opening like' cytoplasmic domain of Igα and Igβ complex during the transmembrane signaling transduction of BCR activation (*Tolar et al., 2005*).

In conclusion, all these results define the sensitivity and threshold for mechanical force that is required to activate IgM-BCRs and isotype-switched IgG-BCR and IgE-BCR, highlighting the significant contribution of mechanical force signals in the enhanced activation of isotype-switched IgG-BCR and IgE-BCR expressing memory B cells in rapid and high-titered antigen recall responses.

## Materials and methods

### Cells, antigens, antibodies and plasmid constructs

Mouse J558L cells stably expressing Igα-YFP and high affinity version of NP-specific B1-8-mIgM-CFP or mIgG-CFP were constructed and maintained as previously described (*Liu et al., 2010a*). J558L B cells and Human Ramos B cells were cultured in RPMI 1640 medium containing 10% FBS, penicillin and streptomycin antibiotics (Invitrogen, Carlsbad, CA) as described (*Liu et al., 2010c*; *Sohn et al., 2011*). Both of these B cell lines were gifts from Dr Susan K Pierce (NIAID, NIH, USA). Primary B1-8 specific mature naive B cells were negatively isolated from spleens of the IgH$^{B1-8/B1-8}$ Igk$^{-/-}$ transgenic mice as reported previously (*Liu et al., 2010a*). Following our published protocol (*Liu et al., 2010b*), the spleen B cells from IgH$^{B1-8/B1-8}$ Igk$^{-/-}$ transgenic mice were incubated with 40 µg/ml LPS and 20 ng/ml recombinant mouse IL-4 (rm-IL-4) for 3 days to induce the isotype-switch to IgG1-BCR.

DyLight 649 AffiniPure Fab Fragment Goat Anti-Mouse IgM, µ Chain Specific (Jackson Immuno-Research, West Grove, PA) and Alexa Fluor 647 AffiniPure Fab Fragment Goat Anti-Mouse IgG (H + L) (Jackson ImmunoResearch, West Grove, PA) were used for IgM and IgG staining accordingly. In short, the cells were stained with 100 nM antibody on ice for 10 min, after washing three times with PBS, the cells are ready for further experiments. FITC-conjugated mouse IgG2a anti-NP antibody named Mouse IgG2a isotype control antibody was purchased from Miltenyi Biotec (Auburn, CA).

## Preparation of B1-8 NP-specific tension gauge tethers (NP-TGTs)

NP-TGTs were prepared following the published protocols (Wang and Ha, 2013) with modifications. To specifically activate B cells, we conjugate NP (4-Hydroxy-3-nitrophenylacety) hapten to one DNA strand (single strand DNA, ssDNA) of the double strand DNA (dsDNA) of TGTs. Briefly, NP-e-Aminocaproyl-OSu (Biosearch Technologies, Petaluma, CA) were conjugated to ssDNA with NH2 group modification. The sequence is as below:

5′-CAC AGC ACG GAG GCA CGA CAC-NH2/-3′

The other strand of the TGT has a biotin tag at a pre-designed position for performing different rupture force points and binding to the coverslip through biotin–neutravidin bond. The sequence is as below:

5′-GTG TCG TGC CTC CGT GCT GTG-3′ with biotin label at position 1, 2, 4, 7, 11, 15, 18, and 21 base, which form 12, 16, 23, 33, 43, 50, 54, and 56 pN, respectively. NP-ssDNA and biotin-ssDNA were further annealed in the annealing buffer following the protocol from Invitrogen.

Coverslip (VWR International) were pretreated with stripping buffer ($H_2SO_4:H_2O_2 = 7:3$), washed and dried before were glued to the disposable 8-well chamber frame (Nunc Lab-Tek chamber). And then 200 µg/ml neutravidin were added to the coverslip, after incubation for 30 min, extensive washing was performed. NP-TGTs were then loaded to the coverslip at the concentration of 50 nM for 30 min at room temperature for the purpose of tethering NP-TGTs to the coverslip. After carefully washing with PBS, the coverslip was blocked with 1% casein (wt/vol) in PBS for 30 min. And then, the NP-TGT conjugated coverslip was ready for use. The cells were then loaded on the surface for reaction at 37°C for 10 min, if no specific indications.

## Preparation of planar lipid bilayers (PLBs)

PLBs were prepared following our published protocol (Liu et al., 2010a; Wan and Liu, 2012), which biotinylated NP8-BSA were attached to biotin lipid through streptavidin. In brief, biotin liposomes were prepared by sonication of 1,2-Dioleoyl-sn-Glycero-3-phosphocholine and 1,2-Dioleoyl- sn-Glycero-3-phosphoethanolamine-cap-biotin (Avanti Polar Lipids, Alabaster, AL) in a 25:1 molar ratio in PBS at a lipid concentration of 2.5 mM. The PLBs were prepared in 8-well chambers (Nunc Lab-Tek) with the coverslip cleaned by nanostrip buffer. The coverslip was incubated with 0.1 mM biotin liposomes in PBS for 20 min at room temperature. After washing with 10 ml PBS, the PLB was incubated with 40 nM streptavidin for 15 min and excessive streptavidin was washed away with 10 ml PBS. And then the streptavidin-containing PLBs were incubated with 30 nM biotinylated, NP8-BSA (pre-mixed Alexa488-conjugated NP8-BSA with non-fluorescent conjugated NP8-BSA at 1:10 molar ratio) for 15 min. After washing, PLBs were blocked with 1% (wt/vol) Casein in PBS for 30 min at 37°C and washed thoroughly for further use.

## Molecule imaging by total internal reflection fluorescence microscope (TIRFM) and confocal fluorescence microscope

TIRFM images were acquired by an Olympus IX-81 microscope equipped with a TIRF port, Andor iXon+ DU-897D electron-multiplying CCD camera, Olympus 100× 1.49 N.A. objective lens. The acquisition was controlled by Metamorph software (Molecular Devices). The excitation area is about 11,600 µm². Laser power was measured at the head of the objective lens by placing the laser beam at a perpendicular orientation to the imaging plane. The laser power of 647 nm laser in BCR imaging is about 7.61 µW (2% output power only in the configuration of Metamorph supported operating panel; 647 nm laser is 10 mW if using 100% output power). The laser power of 488 nm laser for the imaging of NP-TGT sensors by FITC-conjugated NP-specific antibodies is about 22.9 µW (5% output power only in the configuration of Metamorph supported operating panel), and the laser power is about 377.1 µW in single molecule imaging based experiment for the quantification of surface density of NP-TGT sensors (20% output power only in the configuration of Metamorph supported operating panel; 488 nm laser is 3 mW if using 100% output power). For the imaging options, the exposure time was 100 ms for 512 × 512 pixels image, unless specially indicated. All the images are confirmed not over exposed by the software.

Confocal images were acquired by the Olympus FLUOVIEW FV1000 confocal laser scanning microscope with a 100× oil objective lens. The reconstructed side view images were processed by Bitplane Imaris. All the images were analyzed and processed with Image J (NIH, U.S.) software as indicated. The mean fluorescence intensity (MFI) and the total fluorescence intensity (Total FI), as

arbitrary unit, of BCRs and signaling molecules accumulated to the IS were calculated based on the intensity and area analysis as described (*Lakadamyali et al., 2004*; *Liu et al., 2010b*, *2010c*, *2012b*).

## Intracellular immunofluorescence staining and molecules imaging

The recruitment of signaling molecules into the IS of B cells stimulated by NP-TGTs was imaged by TIRFM, and the detection of internalized BCR is imaged by Confocal following our previously published protocol (*Liu et al., 2010a*, *2010b*, *2010c*, *2012b*). In brief, BCRs were pre-stained and then B cells were loaded to the chambers for reaction with NP-TGTs for 10 min followed by 4% paraformaldehyde fixation. After washed with 10 ml PBS, the B cells were permeabilized with 0.1% Triton X-100 and then blocked with 100 μg/ml goat non-specific IgG (Jackson ImmunoResearch Laboratory, West Grove, PA). Subsequently, cells were stained with different primary antibodies including phospho-Zap-70 (Tyr319)/Syk (Tyr352) antibody (Cell Signaling Technology), phospho-PLCγ2 (Tyr759) antibody (BD), phospho-PI3K (Tyr458) antibody (Cell Signaling Technology, Danvers, MA), anti-phosphotyrosine (Millipore Upstate, Billerica, MA) at 37°C for 1 hr. After washed with 10 ml PBS, B cells were stained with secondary antibody Alexa Fluor 568-conjugated F(ab′)$_2$ goat antibody specific for rabbit or mouse IgG (Invitrogen, Carlsbad, CA) as previously described (*Liu et al., 2012b*). Images were analyzed by software Image J (NIH, U.S.) following our published protocols (*Liu et al., 2010a*, *2010b*, *2010c*, *2012b*).

## Treatment of B cells with pharmaceutical inhibitor

For inhibitor studies, primary mature naive B cells from IgH B1-8/B1-8 Igk−/− transgenic mice were pretreated with myosin IIA inhibitor Blebbistatin (*Hung et al., 2013*), FAK inhibitor PF573-228 (Sigma–Aldrich, St. Louis, MO) (*Slack-Davis et al., 2007*), dynein inhibitor HPI-4 (Sigma–Aldrich, St. Louis, MO) (*Firestone et al., 2012*), or internalization inhibitor MDC (Sigma–Aldrich, St. Louis, MO) (*Liu et al., 2010a*) before the imaging experiment. Briefly, B cells were pretreated with 50 μM Blebbistatin at 37°C for 30 min, 1 μM PF573-228 at 37°C for 30 min, 30 μM HPI-4 at 37°C overnight, or 100 μM MDC for 30 min at room temperature to block the function of myosin IIA, FAK, or dynein, respectively. As a control, B1-8 primary B cells were pretreated with DMSO for 30 min at 37°C, or overnight as the control for HPI-4 inhibitor, or 30 min at room temperature as the control for MDC inhibitor.

## Acknowledgements

We thank Dr Susan K Pierce (National Institute of Allergy and Infectious Diseases, National Institutes of Health), Dr Klaus Rajewsky (Immune Regulation and Cancer, Max-Delbrück-Center for Molecular Medicine), and Dr Mark Shlomchik (Yale University) for generously providing experimental materials. We thank Dr Mian Long (Chinese Academy of Sciences), Dr Hai Qi (Tsinghua University), Dr Yujie Sun (Peking University) and Dr Wei Chen (ZheJiang University) for helpful discussions. This work is supported by funds from Ministry of Science and Technology of China (2014CB542500, 2014AA020527), National Science Foundation China (81361120384, 31270913 and 81422020), Beijing Natural Science Foundation (5132016), One-Thousand-Youth-Talents program (2069999-3) of Chinese Central Government and Tsinghua University Initiative Scientific Research Program (20131089279).

## Additional information

### Funding

| Funder | Grant reference | Author |
| --- | --- | --- |
| Ministry of Science and Technology of the People's Republic of China | 2014CB542500 | Wanli Liu |
| Ministry of Science and Technology of the People's Republic of China | 2014AA020527 | Wanli Liu |
| National Science Foundation China | 81361120384, 31270913 and 81422020) | Wanli Liu |
| Beijing Natural Science Foundation | 5132016 | Wanli Liu |

| Funder | Grant reference | Author |
| --- | --- | --- |
| Chinese Central Government One-Thousand-Youth-Talents program | 2069999-3 | Wanli Liu |
| Tsinghua University Initiative Scientific Research Program | 20131089279 | Wanli Liu |

The funders had no role in study design, data collection and interpretation, or the decision to submit the work for publication.

## Author contributions

ZW, Conception and design, Acquisition of data, Analysis and interpretation of data, Drafting or revising the article; XC, Experimental material providing, Conception and design; HC, Experimental material providing, Acquisition of data, Analysis and interpretation of data; QJ, Technical support, Analysis and interpretation of data, Drafting or revising the article; YC, Critical experiment performing, Acquisition of data, Analysis and interpretation of data; JW, YC, Experimental material, Conception and design; FW, Giving critical revision, Drafting or revising the article; JL, Technical support and giving critical revision, Drafting or revising the article; ZT, Experimental material, Acquisition of data, Analysis and interpretation of data; WL, Conception and design, Drafting or revising the article

## Additional files

### Supplementary file

• Source code 1. Matlab supported 2D Gaussian fitting code.

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
