## [Decision Letter]

Thank you for sending your work entitled “Activation of IgM- or isotype-switched IgG- and IgE-B cell receptor requires distinct single molecular force threshold” for consideration at *eLife*. Your article has been favorably evaluated by Michael Marletta (Senior Editor), a Reviewing Editor, and three reviewers.

The following individuals responsible for the peer review of your submission have agreed to reveal their identity: Michael Dustin and Matthew Lang (peer reviewers).

This is a potentially important study that utilizes force-calibrated DNA duplexes to determine if there is a threshold for B cell receptor and associated activated signaling molecules to be recruited to a planar substrate. One strand of the duplex is attached to the substrate and the other strand has a hapten that is recognized by an IgM or IgG on the surface of a live B cell. The duplexes are claimed to be calibrated to dissociate to single strands with application of forces exceeding a threshold in the range of 12-56 pN, which is a biologically relevant range. When dissociation takes place the strand with the hapten would go with the BCR, but because the engagement is monovalent it would likely lose any ability to hold the receptor in the interface or stimulate signaling. The authors follow accumulation of BCR in the interface and recruitment of phosphorylated signaling molecules that associate with the BCR. They find that IgM generates a continuum of responses across the range of force thresholds, whereas IgG activates similarly across the entire spectrum. They show that this difference in sensitivity to the linker is encoded by the short cytoplasmic domains of IgM and IgG heavy chains based on analysis of chimeric receptors. The paper contains important quantitative new insights into mechanisms of B cell receptor signaling and the effects of heavy chain isotype on this process, but a number of issues must be resolved before the work is suitable for publication in *eLife*:

1) It is important to determine if what the authors are looking at is actually mechanotransduction, or rather an effect related to the amount of force the receptor can exert on the substrate to extract antigen. This issue was recently addressed qualitatively by Natkanski et al., which is in the reference list, but not cited or discussed adequately in the text. Addressing this issue quantitatively would be a significant advance.

The molecular mechanism of the postulated mechanosensitivity is left unanswered (the authors exclude a major involvement of integrin signaling, dynein or myosin motor activity; these experiments need controls that the inhibitors are working). Further, the authors use total internal reflection fluorescence microscopy to focus on the interface. This is a powerful method for quantification of interactions in interfaces, but can miss important information about the rest of the cell. It would be important to verify whether or not the IgM receptors that break the weaker duplexes just remain at the surface or if they are internalized and removed from the cell surface. This would be readily evaluated by confocal microscopy in permeabilized cells. This would provide an important denominator for the analysis of the BCR in the interface. If the weaker duplexes actually down-regulate the IgM from the surface this would be very different from simply not engaging or activating receptors. This is important to the conclusions of the study, which could go in one of two directions: the issue of antigen harvesting from the surface vs. mechanotransduction by the receptor. These are not mutually exclusive or necessarily related.

This same issue could alternatively be approached experimentally by employing fluorescently labeled antigen-containing strands in the tethers, although this would require production of new, more expensive tethers. The potential challenge with this approach is that the single Ig-hapten interaction is short lived and the DNA may dissociate from the Ig before it can be internalized. But if it is unlikely to re-anneal to the glass surface following tether failure, the cell might create a hole in a substrate that is undergoing many failure events. If this doesn't happen then the case for mechanotransduction would be stronger. In this regard, Figure 1 done in the presence of cells with NP-specific BCR, or probes lacking NP, but containing a fluorophore reporter, would also be helpful. It would be good to see that areas of the surface contacting cells were the same brightness as locations not containing cells if mechanosensing and not antigen scavenging is at work.

2) A puzzling issue in this experimental system is that the monovalent NP hapten coupled to the DNA probes should dissociate from the B1-8 BCR with a half-life of about 1 second and withstand forces of approximately 10-15 pN (see Natkanski et al., Science 2013). This is about as much force as is needed to separate the weakest tethers in this manuscript, and much less than the separation forces of the tethers at the two thresholds for B cell spreading (23 pN and 50 pN). The two thresholds correspond to approximately two times and four times the rupture force of the single NP-B1-8 bond, raising a possibility that the separation somehow depends on multivalent bonds. If the aim of the manuscript is to be quantitatively precise about the single molecule forces, this should be addressed.

The experimental setup of the authors differs from the Ha paper in that the tethers are bound to glass-absorbed neutravidin, instead of neutravidin bound to a biotin-PEG covalently attached to the glass. It is thus possible that some neutravidin (which binds quite weakly to glass) dissociates under forces applied through multiple tethers. It is also possible that some of the tethers bind directly to glass and cannot be separated. This could be investigated by controls without neutravidin and using fluorescent neutravidin to measure its release from underneath the cells.

Finally, it should be discussed whether the differences observed with different tethers could be not just because of mechanosensitivity of the BCR, but also because of the half-life of the bonds of the BCR with the substrate, since sensor rupture will terminate BCR signalling similarly as does BCR dissociation from the antigen.

3) There are serious concerns with the way in which the paper describes the tethers and also with the specific threshold forces ascribed to each sensor.

In many cases the authors refer to the ligands as a force gauge. A ‘gauge’ implies multiple calibrated levels. While the panel of ligands as a whole may represent a gauge, individually they are threshold sensors, acting as binary and not continuous indicators.

The ligands were derived from work in reference [69]. They are treated in this manuscript as having specific force thresholds [12, 16, 23, 33, 43, 50, 54 and 56 pN]. These are double strands of DNA with various lengths, compositions and pulling geometries, shearing and unzipping. Interpreting them as having specific force thresholds is misleading. The rupture force of these types of systems has been investigated extensively and depends on loading rate or time under load. These are estimated force ranges, based on a 2 second load. The experiments here are on the timescale of 10 minutes, and thus very different forces from the ones quoted in the bracketed numbers above may actually be delivered. The authors need to calibrate the probes under their conditions, providing force ranges +/- for each of the probes listed (see Lang et al., Nature Methods 2004 Figure 5, where similar unzipping and shearing geometries were measured and quantified for dsDNA linkage system).

It is also important to note that the FWHM for unzipping spans 5 pN and shearing geometry 15 pN. Thus when the authors make fine distinctions about the force probes as listed above, the actual range of any given force probe contains a spread overlapping other probe ranges. This spread may influence interpretation and the readers need to know that these probes provide a range of forces. Further, the experimental conditions here would shift this whole panel to lower forces as the loading times are orders of magnitude longer than the design times by Ha (2 seconds). Statements in the text such as “precisely defined single molecular forces” are hence misleading.

The BCRs have two binding sites for each transmembrane linkage (Figure 6). Is the surface density of probes low enough such that only one is engaged? Otherwise if two nominally 12 pN probes bind mIgG, the total force can reach 24 pN.

What is the surface density of ligands in these experiments in molecules/µm^2^? Experiments were “repeated using a 10-fold lower concentration” with the same observed response. If both concentrations are still saturating then this control is not valid. The surface density could easily be estimated using FITS labeled Figure 1 control structure. Supplemental figure 4 shows 50nM, 10nM and 2nM concentrations, yet the total fluorescence does not drop by these same values. The surface density of ligands should be measured independently and quantified.

Results in Natkanski will help the authors place bounds on expected timescales, loading rates etc. when mapping onto their system. The AFM in Natkanski involves a higher loading rate, stiffer spring, and shorter lifetime situation than here (although the coverglass is stiff, the cell is not). Based on the Natkanski result, the relevant timescale in the present experiments is likely seconds to tens of seconds and not minutes. This prior paper should certainly be worked into the Discussion.

---

## [Author Response]

*1) It is important to determine if what the authors are looking at is actually mechanotransduction, or rather an effect related to the amount of force the receptor can exert on the substrate to extract antigen. This issue was recently addressed qualitatively by Natkanski et al., which is in the reference list, but not cited or discussed adequately in the text. Addressing this issue quantitatively would be a significant advance*.

We agree. We paid great attention to this point when revising our manuscript. We are fully aware of the relevance of the published studies by Natkanski et al., (Science, 2013, 340:1587-90) to our results. In addition to what we have referred to in our original manuscript, we further cited and discussed the following key points from Natkanski’s paper in our revised manuscript: (1) Internalization of antigen from different types of substrates; (2) time scale of the mechanical force under load; (3) rupture force profiles between B1-8-BCR and NP antigen.

*The molecular mechanism of the postulated mechanosensitivity is left unanswered (the authors exclude a major involvement of integrin signaling, dynein or myosin motor activity; these experiments need controls that the inhibitors are working). Further, the authors use total internal reflection fluorescence microscopy to focus on the interface. This is a powerful method for quantification of interactions in interfaces, but can miss important information about the rest of the cell. It would be important to verify whether or not the IgM receptors that break the weaker duplexes just remain at the surface or if they are internalized and removed from the cell surface. This would be readily evaluated by confocal microscopy in permeabilized cells. This would provide an important denominator for the analysis of the BCR in the interface. If the weaker duplexes actually down-regulate the IgM from the surface this would be very different from simply not engaging or activating receptors. This is important to the conclusions of the study, which could go in one of two directions: the issue of antigen harvesting from the surface vs. mechanotransduction by the receptor. These are not mutually exclusive or necessarily related*.

Thank you for all these insightful suggestions. We provide the positive control experiments for each inhibitor to show that these inhibitors are working in pretreated B cells (please refer to Figure 5—figure supplement 1). We agree with the reviewers on the importance of evaluating through confocal microscopy the internalization of BCR and antigen molecules after B cell activation. All these experiments can better help us understand the mechanisms of the different response of B1-8-BCRs when encountering the mechanical forces from NP-TGT sensors. We performed these experiments and the results showed that there is no obvious internalization of BCRs in case of each of these three NP-TGT sensors (12, 43 and 56 pN, see Figure 2—figure supplement 2). We did not detect obvious internalization events at time point of 10 min after the immune recognition of B1-8-BCR and NP-TGT (Figure 2—figure supplement 2). We picked up a time point of 10 min to examine the efficiency of internalization as we used 10 min as an end point to compare the different activation of B cells when encountering NP-TGT sensors providing low-, middle- or high-forces in our original and revised manuscript. All these experiments demonstrated that the different levels of the synaptic accumulation of BCRs are mainly induced by the different mechanical forces applied to BCRs through NP-TGT sensor, rather than induced by the antigen acquisition and the subsequent internalization events. To further confirm this statement, we pretreated B cells with monodansylcadaverine (MDC), which is an inhibitor to block internalization (Tolar P, et al., Nature Immunology, 2005, 6:1168-76; Chaturvedi A, et al., Immunity, 2008, 28:799-809; Liu W, et al., The Journal of Experimental Medicine, 2010, 207:1095-111). We found the activation of the pretreated B cells still maintained the sensitivity to the mechanical forces and exhibited the multi-threshold effect (Figure 2—figure supplement 2). Thus, all these new data showed that, regarding “the issue of antigen harvesting from the surface vs. mechanotransduction by the receptor*”*, mechanical force mainly drives the different response in the initiation of B cell activation.

We also discussed in the revised manuscript that the lack of internalization of BCR within 10 min of the recognition of NP-TGT antigens presented on the surface of coverslip is consistent with the published studies by Natkanski et al. (Science, 2013, 340:1587-90) and Fleire et al. (Science, 2006, 312:738-41). Natkanski et al. reported that soluble antigen or antigen presented by plasma membrane sheets (PMS) but not planar lipid bilayers (PLBs) can support the Myosin IIA mediated internalization of BCRs and antigens at an end point of 20 min after the immune recognition (Science, 2013, 340:1587-90). Fleire et al. can only examine and quantify the efficiency of the internalization of antigen at an end point as early as 2 hrs after placing the B cells on PLBs containing different densities of antigen (20, 100, and 200 molec/µm^2^) (Science, 2006, 312:738-41). Both of these studies indicate that B cells can only efficiently internalize BCRs and antigens on softer substrates (like PMS) in short period of incubation time (20 min) or on stiffer substrates but with very long incubation time (2 hrs). In our experimental condition (10 min reaction time and the highest density of the antigen at 29 molec/µm^2^ on stiff coverslip), it is not a surprise that internalization would only occur at a very low efficiency.

*This same issue could alternatively be approached experimentally by employing fluorescently labeled antigen-containing strands in the tethers, although this would require production of new, more expensive tethers. The potential challenge with this approach is that the single Ig-hapten interaction is short lived and the DNA may dissociate from the Ig before it can be internalized. But if it is unlikely to re-anneal to the glass surface following tether failure, the cell might create a hole in a substrate that is undergoing many failure events. If this doesn't happen then the case for mechanotransduction would be stronger. In this regard,*
Figure 1
*done in the presence of cells with NP-specific BCR, or probes lacking NP, but containing a fluorophore reporter, would also be helpful. It would be good to see that areas of the surface contacting cells were the same brightness as locations not containing cells if mechanosensing and not antigen scavenging is at work*.

We agree with the reviewer that additionally there are two alternative methods to address the internalization question as discussed above: (1) to use a fluorophore-conjugated version of NP-TGT, for example, Cy3-NP-TGT; (2) to use NP-specific antibody to stain the quantity of the NP-TGTs on the surface of the coverslip. When revising the manuscript, we chose the second approach to re-address the internalization question. However we found that it is technically difficult to accurately stain the NP-TGT sensors by NP-specific antibodies at the interface between the B cells and coverslip presenting high-force NP-TGT (this interface was termed as B cell IS thereafter). In our tests, we performed the standard protocol to stain NP-TGT sensors in permeabilized B cells following the published studies of ours and those of others (Vascotto F, et al., The Journal of Cell Biology, 2007, 176:1007-19; Liu W, et al., The Journal of Experimental Medicine, 2010, 207:1095-111; Kumari S, et al., *eLife*, 2015, v.4; Tolar P, et al., Immunity, 2009, 30:44-55). We cannot detect obvious fluorescent signal for 56 pN NP-TGT sensor (fluorophore-conjugated NP-specific antibodies) at the area within the B cell IS, although such signal can be readily detected at the area without B cells (see Figure 8). We tested different batches of NP-specific polyclonal antibodies and consistently observed these results. Although as the reviewers have commented that “…the cell might create a hole in a substrat…,” we speculate there are two explanations for the production of these “holes” (area lack of fluorescent signal, see Figure 8): (1) NP-conjugated DNA strain on 56 pN NP-TGT sensors were ruptured and dissociated from the coverslip. Subsequently, these dissociated NP-DNA strains can either be internalized by the B cells or diffuse into the bulky aqueous buffer; (2) The adhesion of B cells with 56 pN NP-TGT presenting coverslip was so tight that it would greatly diminish the accessibility of NP-specific antibodies to NP-TGT molecules within the B cell IS. Through a series of further tests, we figured out that the second possibility makes more sense.

Author response image 1.**DOI:**
http://dx.doi.org/10.7554/eLife.06925.021

First, we used interference reflection microscopy (IRM) to image B cell IS and showed the strong adhesion between B cells and the 56 pN NP-TGT-containing coverslip (see Figure 8). These dark IRM images usually suggest a very tight contact with a gap of less than 15 nm (Barr VA, et al., Current Protocols in Cell Biology, 2009, Chapter 4: Unit 4.23). The observation of the tight contact of B cells with antigen-presenting coverslip were consistent with the published studies of ours and those of others (Xu L, et al., Journal of Leukocyte Biology, 2015, doi:10.1189/jlb.2A0614-287RR; Liu C, et al., The Journal of Immunology, 2012, 188:3237-46; Seeley-Fallen MK, et al., Proceedings of the National Academy of Sciences, 2014, 111:9881-6). Second, to prevent the possibility that the lack of potential interaction of NP-TGT with NP-specific antibodies was due to the low stoichiometry of NP in NP-TGT molecules, we used another antigen with high NP stoichiometry, NP32-BSA, for a double check. We are able to confirm that NP-specific antibodies cannot efficiently stain the NP32-BSA molecules within the B cell IS (see Figure 8).

Thus, it is technically difficult to accurately stain the high-force NP-TGT sensors by NP-specific antibodies within the B cell IS. This technical difficulty would prevent us from using this alternative approach to assess the effects of internalization in the different levels of BCR accumulation within the B cell IS during B cell activation that were induced by low-, middle- or high-mechanical force NP-TGT molecules. Since we have used the non-alternative method as recommended by the reviewer to address this question, we will not include these negative NP-TGT staining data (Figure 8) in our manuscript.

*2) A puzzling issue in this experimental system is that the monovalent NP hapten coupled to the DNA probes should dissociate from the B1-8 BCR with a half-life of about 1 second and withstand forces of approximately 10-15 pN (see Natkanski et al., Science 2013). This is about as much force as is needed to separate the weakest tethers in this manuscript, and much less than the separation forces of the tethers at the two thresholds for B cell spreading (23 pN and 50 pN). The two thresholds correspond to approximately two times and four times the rupture force of the single NP-B1-8 bond, raising a possibility that the separation somehow depends on multivalent bonds. If the aim of the manuscript is to be quantitatively precise about the single molecule forces, this should be addressed*.

The reviewers raised an important suggestion. We would definitely agree with the reviewers that our experimental system does not allow us to make the statement that we are precisely quantifying single molecule force, or more specifically single bond rupture, during the activation of BCRs. A major reason is that BCR monomer itself is a well-characterized bivalent binding receptor with two identical antigen binding domain (Schamel, W.W., and Reth, M., Immunity, 2000, 13:5-14.). Additionally, the recent advance of the super-resolution imaging studies demonstrated that BCR forms obvious nanoclusters with 20-50 IgM-BCRs per nanocluster on quiescent B cells (Mattila PK, et al., Immunity, 2013, 38:461-74). Moreover, the time scale we used in our experiments also indicated that it might not be a monovalent bond. All these points demonstrated the high possibility of multivalent bonds based mechanical forces in regulating the initiation of B cell activation. In the revised manuscript, we eliminated the overstatement of “single molecule force” or “single-bond rupture”, but rather reinforce the concept of how mechanical forces applied to BCRs by the NP-TGT antigens influence B cell activation.

In addition, we would like to share with the reviewers our thoughts that direct comparison between the rupture force of TGT sensors (Wang X and Ha T, Science, 2013, 340:991-4) and the rupture force between NP hapten antigen and B1-8 BCR (Natkanski et al., Science, 2013, 340:1587-90) should be avoided. The conditions where the rupture force values were obtained are different in these two papers. The values of the rupture forces of these TGT sensors were calculated based on the data measured by magnetic tweezers system with a loading rate of 0 pN/s (constant force with a time under force mode of 2 sec during the measurements), while the values of the rupture forces between NP hapten antigen and B1-8 BCR were measured by AFM system with a loading rate of around 1000 pN/s if we refer to a rupture force of 12-15 pN (different loading rates will lead to the calculation of quite different rupture forces in AFM measurements). Rupture forces measured at different conditions (different loading rate, time under load, spring constants, etc.) in even the same hardware system (either magnetic tweezers or AFM system) are different. Also quite different is the sensitivity for the measurement of mechanical forces of these two hardware systems. Thus, in our opinion, the mechanical force values of the TGT sensors and NP-antigen/BCR cannot be compared directly. In the revised manuscript, we discussed that it would be of great importance to use the magnetic tweezers experimental system to examine the interaction of B1-8-IgM-BCR with NP antigen, and ask how the magnitude and duration of mechanical forces will influence B cell activation.

*The experimental setup of the authors differs from the Ha paper in that the tethers are bound to glass-absorbed neutravidin, instead of neutravidin bound to a biotin-PEG covalently attached to the glass. It is thus possible that some neutravidin (which binds quite weakly to glass) dissociates under forces applied through multiple tethers. It is also possible that some of the tethers bind directly to glass and cannot be separated. This could be investigated by controls without neutravidin and using fluorescent neutravidin to measurement its release from underneath the cells*.

We concur with the concerns from the reviewer that the neutravidin may dissociate from coverslip over time and that the non-specific tethering of NP-TGT to neutravidin-blank coverslip may occur. In the revised manuscript, we investigated the dissociations of neutravidin from the coverslip within the maximal time course of our experiments by using fluorophore-conjugated neutravidin. No obvious dissociation of the neutravidin was detected in a 10 min incubation time course (see Figure 1—figure supplement 1). We also investigated the non-specific binding of NP-TGTs to the coverslip without the pre-coated neutravidin (see Figure 1—figure supplement 1). Lastly, we confirm that B cells cannot be activated by those potential NP-TGT sensors that are attached to the coverslip without pre-coated neutravidin (see Figure 1—figure supplement 1).

*Finally, it should be discussed whether the differences observed with different tethers could be not just because of mechanosensitivity of the BCR, but also because of the half-life of the bonds of the BCR with the substrate, since sensor rupture will terminate BCR signalling similarly as does BCR dissociation from the antigen*.

We fully agree. We discussed in the revised manuscript that lifetime between BCR and antigen could also contribute to the observed difference of these NP-TGT sensors in initiating B cell activation. In the Discussion, we cited the recent observation of how bond lifetime between TCR and pMHC is demonstrated to dramatically affect TCR signaling through a catch bond manner (Liu B, Chen W, et al., Cell, 2014, 157:357-68).

*3) There are serious concerns with the way in which the paper describes the tethers and also with the specific threshold forces ascribed to each sensor*.

*In many cases the authors refer to the ligands as a force gauge. A ‘gauge’ implies multiple calibrated levels. While the panel of ligands as a whole may represent a gauge, individually they are threshold sensors, acting as binary and not continuous indicators*.

Thank you very much for the corrections. We made the changes in the revised manuscript.

*The ligands were derived from work in reference*
[69]*. They are treated in this manuscript as having specific force thresholds [12, 16, 23, 33, 43, 50, 54 and 56 pN]. These are double strands of DNA with various lengths, compositions and pulling geometries, shearing and unzipping. Interpreting them as having specific force thresholds is misleading*.

We agree. We specifically indicated these facts in the revised manuscript. In particular, we stated that the specific force value is not an absolute value but presents a distribution with a FWHM of 5 pN for unzipping rupture mode and 15 pN for shearing rupture mode (Lang MJ, et al., Nature Methods, 2004, 1:133-9). In the revised manuscript, we discussed that the actual range of any given TGT sensors shall not be fixed but can be best represented by an overlapped region by the nearby TGT sensors. We also discussed how these features may help to interpret the multi-threshold effect of mechanical forces on the initiation of BCR activation in the revised manuscript.

*The rupture force of these types of systems has been investigated extensively and depends on loading rate or time under load. These are estimated force ranges, based on a 2 second load. The experiments here are on the timescale of 10 minutes, and thus very different forces from the ones quoted in the bracketed numbers above may actually be delivered. The authors need to calibrate the probes under their conditions, providing force ranges +/- for each of the probes listed (see Lang et al., Nature Methods 2004*
Figure 5*, where similar unzipping and shearing geometries were measured and quantified for dsDNA linkage system)*.

We appreciated all these suggestions and advice. However, we would like to clarify that the time scale of rupturing any single NP-TGT sensor in our experiments system is not 10 minutes. The 10 minute time point is just the end point that we stopped the interaction of the B cell with NP-TGT sensors and used TIRFM imaging to quantify the strength of B cell activation. In our original manuscript, we also used time lapse TIRFM imaging to examine the dynamics in a 10 minute time course of the response of B cells when initially encountering NP-TGT molecules (Figure 2) and we can conclude that the high-force NP-TGT sensors obviously drove better B cell activation responses than low-force NP-TGT ones as early as 3 minutes after loading the B cells to NP-TGT presenting coverslip. We clarified this point in the revised manuscript to avoid confusions. Then what should the time scale of rupturing a single NP-TGT sensor at the physiological condition be? We have to say that there is no direct experimental evidence to allow us to tell the exact answer. We fully agree with the reviewer that the results in the paper by Natkanski et al. can best help us answer this question. We also agree with the reviewer that the best guess of the relevant time scale of rupturing single NP-TGT sensor in our experimental system would be a seconds to dozens of seconds. From this point of view, such a time scale would definitely be larger but does not differ too much from the 2-sec force under load time scale that was used to measure the rupture force of TGT molecules by Ha and his colleagues (Wang X and Ha T, Science, 2013, 340:991-4).

Lastly, optical tweezers or magnetic tweezers should be used to acquire the spectrum with FWHM value of each NP-TGT molecule, using a time under load value that we need to subjectively choose from seconds to dozens of seconds. As a lab doing conventional immune cell biology studies, we lack the expertise to set up such a tweezers system to do these experiments, and it was extremely difficult for us to find local collaborators that could finish this experiment in the requested amount of time. However, we did take the suggestions from the reviewers to extensively discuss these points in the revised manuscript. In particular, we remind the reviewers that there is no direct evidence to show that the absolute value of the mechanical forces that were taken from the TGT paper (Wang X and Ha T, Science, 2013, 340:991-4) can exactly fit to the situation between B1-8-BCR and NP antigen at the physiological condition. Thus, the usage of low-force NP-TGT, middle-force NP-TGT and high-force NP-TGT might better represent the core findings of this manuscript that the activation of IgM-BCR and isotype-switched BCR exhibit distinct sensitivity and threshold toward the mechanical forces. We did make changes in the revised manuscript to reflect all these facts. Lastly, in our opinion, although the absolute values of the mechanical forces of the TGT sensors used in our experimental condition might differ from those in the paper of Ha and his colleagues (Wang X and Ha T, Science, 2013, 340:991-4), the trends of the dependence on high, middle or low mechanical forces should be the same.

*It is also important to note that the FWHM for unzipping spans 5 pN and shearing geometry 15 pN. Thus when the authors make fine distinctions about the force probes as listed above, the actual range of any given force probe contains a spread overlapping other probe ranges. This spread may influence interpretation and the readers need to know that these probes provide a range of forces. Further, the experimental conditions here would shift this whole panel to lower forces as the loading times are orders of magnitude longer than the design times by Ha (2 seconds). Statements in the text such as “precisely defined single molecular forces” are hence misleading*.

All are wonderful suggestions. As in our reply to point 2, we took all these suggestions from the reviewers when revising our manuscript. Also, we eliminated the inappropriate language such as “precisely defined single molecular forces”.

*The BCRs have two binding sites for each transmembrane linkage (*Figure 6*). Is the surface density of probes low enough such that only one is engaged? Otherwise if two nominally 12 pN probes bind mIgG, the total force can reach 24 pN*.

*What is the surface density of ligands in these experiments in molecules/µm*^*2*^*? Experiments were “repeated using a 10-fold lower concentration” with the same observed response. If both concentrations are still saturating then this control is not valid. The surface density could easily be estimated using FITS labeled*
Figure 1
*control structure. Supplemental figure 4 shows 50nM, 10nM and 2nM concentrations, yet the total fluorescence does not drop by these same values. The surface density of ligands should be measured independently and quantified*.

Indeed, BCR is a receptor with bivalent binding capability. We took the suggestion of the reviewers to quantify the density (number of molecules/µm^2^) of NP-TGT sensors on the coverslip following a protocol in our previous studies (Liu W, et al., The Journal of Experimental Medicine, 2010, 207:1095-111). For NP-TGT, the incubation concentration of 2 nM, 5 nM, 10 nM and 50 nM resulted in the surface density of 0.3, 4.0, 19.0 and 29.0 per µm^2^ respectively (Figure 2—figure supplement 1). All these data were included into the revised manuscript.

*Results in Natkanski will help these authors place bounds on expected timescales, loading rates etc. when mapping onto their system. The AFM in Natkanski involves a higher loading rate, stiffer spring, and shorter lifetime situation than here (although the coverglass is stiff, the cell is not). Based on the Natkanski result, the relevant timescale in the present experiments is likely seconds to tens of seconds and not minutes. This prior paper should certainly be worked into the Discussion*.

We fully agree with the reviewer on these comments. Our replies to this last point were given in detail when addressing the point regarding calibrating probes above.